# Epistemic Uncertainty Estimation in Regression Ensemble Models with Pairwise Epistemic Estimators

**Lucas Berry, David Meger**
Department of Computer Science
McGill University
lucas.berry@mail.mcgill.ca

## Abstract

This work introduces a novel approach, Pairwise Epistemic Estimators (PairEpEsts), for epistemic uncertainty estimation in ensemble models for regression tasks using pairwise-distance estimators (PaiDEs). By utilizing the pairwise distances between model components, PaiDEs establish bounds on entropy. We leverage this capability to enhance the performance of Bayesian Active Learning by Disagreement (BALD). Notably, unlike sample-based Monte Carlo estimators, PairEpEsts can estimate epistemic uncertainty up to 100 times faster and demonstrate superior performance in higher dimensions. To validate our approach, we conducted a varied series of regression experiments on commonly used benchmarks: 1D sinusoidal data, *Pendulum*, *Hopper*, *Ant*, and *Humanoid*, demonstrating PairEpEsts' advantage over baselines in high-dimensional regression active learning.

## 1 Introduction

In this paper, we propose Pairwise Epistemic Estimators (PairEpEsts) as a non-sample based method for estimating epistemic uncertainty in deep ensembles with probabilistic outputs for regression tasks. Epistemic uncertainty, often distinguished from aleatoric uncertainty, reflects a model's ignorance and can be reduced by increasing the amount of data available [26, 16, 29]. The significance of epistemic uncertainty is particularly pronounced in safety-critical systems, where a single erroneous prediction could lead to catastrophic consequences [43]. Moreover, leveraging epistemic uncertainty proves beneficial as an acquisition criterion for active learning strategies [27].

Previously Monte Carlo (MC) methods have been employed for estimating epistemic uncertainty, in regression tasks, due to the absence of closed-form expressions in most modeling scenarios [15, 4]. However, as the output dimension increases, these MC methods require a large number of samples to get accurate estimates. PairEpEsts leverage Pairwise-Distance Estimators (PaiDEs), which provide a non-sample-based alternative for estimating information-theoretic criteria in ensemble regression models with probabilistic outputs [35].

Ensembles can be seen as committees, with each component acting as a member [50]. PairEpEsts synthesize the consensus amongst a committee of probabilistic learners by calculating the distributional distance between each pair of committee members. Aggregating these distances allows accurate estimation of the mutual information between the model output and weight distribution. When pairwise distances are efficiently computable, PairEpEsts offer a fast, sample-free method to estimate epistemic uncertainty. Figure 1 shows committee members in agreement with low uncertainty (small distance) and disagreement with high uncertainty (large distance).

This study demonstrates the use of PairEpEsts to estimate epistemic uncertainty in regression ensembles with probabilistic outputs. Unlike classification, regression poses unique challenges because entropy is harder to estimate for mixtures of continuous distributions than for categorical

ones. To address this, we focus on Normalizing Flows (NFs), which can capture heteroscedastic and multimodal aleatoric uncertainty [49, 33]. This capability is particularly relevant for robotic locomotion, where nonlinear stochastic dynamics make data acquisition costly and active learning essential. The proposed framework extends epistemic uncertainty estimation to high-dimensional regression tasks that have been largely underserved in the literature. Our contributions are threefold:

- We introduce the framework PairEpEsts, which applies PaiDEs to estimate epistemic uncertainty in deep ensembles with probabilistic outputs (Section 4).
- We extend previous epistemic uncertainty estimation methods to settings with higher-dimensional output [4], and demonstrate how PairEpEsts outperform MC and other active learning baselines in these settings with rigorous statistical testing (Section 5).
- We provide an analysis of the time-saving advantages offered by PairEpEsts compared to MC estimators for epistemic uncertainty estimation (Section 5.4).

## 2 Problem Statement

Following a supervised learning framework for regression, let $\mathcal{D} = \{x_i, y_i\}_{i=1}^N$ denote a dataset, where $x_i \in \mathbb{R}^n$ and $y_i \in \mathbb{R}^d$, and our objective is to approximate a complex multi-modal conditional probability $p(y|x)$. Let $f_\theta(y; x)$ denote our approximation to the conditional probability density, where $\theta$ is a set of parameters to be learned and is distributed as $p(\theta)$.

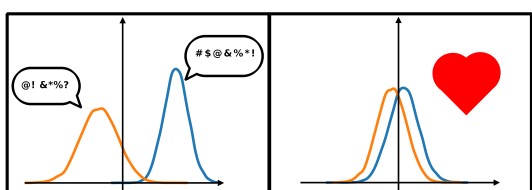

Leveraging a learned model, $f_\theta(y; x)$, we wish to estimate uncertainty which is typically viewed from a probabilistic perspective [14, 29]. When capturing uncertainty in supervised learning, one common measure is that of conditional differential entropy,

Figure 1: Epistemic uncertainty in an ensemble of probabilistic learners is high when the mixture components disagree (left) and low when there is agreement (right).

$$H(y|x) = -\int p(y|x) \ln p(y|x) dy.$$

Utilizing conditional differential entropy, we can establish an estimate for epistemic uncertainty as introduced by Houlsby et al. [27], expressed as:

$$I(y, \theta|x) = H(y|x) - H(y|x, \theta), \tag{1}$$

where $I(\cdot)$ refers to mutual information (MI). Equation 1 demonstrates that epistemic uncertainty, $I(y, \theta|x)$, can be represented by the difference between total uncertainty, $H(y|x)$, and aleatoric uncertainty, $H(y|x, \theta)$. MI measures the information gained about one variable by observing another.

To enable our methods to capture epistemic uncertainty, we employ ensembles to create $p(\theta)$. Ensembles leverage multiple models to obtain the estimated conditional probability by weighting the output distribution from each ensemble component,

$$f_\theta(y; x) = \sum_{j=1}^M \pi_j f_{\theta_j}(y; x) \qquad \sum_{j=1}^M \pi_j = 1, \tag{2}$$

where $M, 0 \le \pi_j \le 1$ and $\theta_j$ are the number of model components, the component weights and the parameters for the $j$th component, respectively. In order to create an ensemble, one of two ways is typically chosen: randomization [5] or boosting [20]. While boosting has led to widely used machine learning methods [10], randomization has been the preferred method in deep learning [39].

Utilizing our ensemble we can estimate epistemic uncertainty in decision-making scenarios such as active learning. When all components produce the same $f_{\theta_i}(y, x)$, $I(y, \theta|x)$ is zero, indicating no epistemic uncertainty. Conversely, when the components exhibit varied output distributions, epistemic uncertainty is high. In an active learning framework, one chooses, at each iteration, what data points to add to a training dataset such that the model's performance improves as much as possible [41, 52].

In our context, one chooses the $x$'s that maximize Equation 1 and adds those data points to the training set as in Bayesian Active Learning by Disagreement (BALD) [27]. It's worth noting that in the realm of continuous outputs as in regression and ensemble models, Equation 1 often lacks a closed-form solution, as the entropy of most mixtures of continuous distributions cannot be expressed in closed form [35]. Hence, prior methods have resorted to MC estimators for the estimation of epistemic uncertainty [15, 48]. One such MC method samples $K$ points from the model, $y_j \sim f_\theta(y; x)$, and then estimates the total uncertainty,

$$\hat{H}_{MC}(y|x) = \frac{-1}{K} \sum_{j=1}^{K} \ln f_\theta(y_j; x).$$

The MC approach approximates the intractable integral over the continuous domain requiring only point-wise evaluation of the ensemble and its components. However, as the number dimensions increases, MC methods typically require a greater number of samples which creates a greater computational burden [51]. Note that the aleatoric uncertainty can be analytically computed as the average of the entropies of each ensemble component distribution.

## 3 Pairwise-Distance Estimators

Unlike MC methods, PaiDEs eliminate sampling by using generalized distance functions between ensemble components. They estimate the entropy of mixture distributions when pairwise distances have closed-form expressions, enabling estimation of $H(y|x)$ from Equation 1. We extend PaiDEs, from Kolchinsky and Tracey [35], to supervised learning and epistemic uncertainty estimation.

### 3.1 Properties of Entropy

One can treat a mixture model as a two-step process: first, a component is drawn and, second, a sample is taken from the corresponding component. Let $p(y, \theta|x)$ denote the joint of our output and model components given input $x$,

$$p(y, \theta|x) = p(\theta_j|x)p(y|\theta_j, x) = \pi_j p(y|\theta_j, x).$$

Using this representation, following principles of information theory [14], we can write its entropy as,

$$H(y, \theta|x) = H(\theta|y, x) + H(y|x). \tag{3}$$

Additionally, one can show the following bounds for $H(y|x)$,

$$H(y|\theta, x) \leq H(y|x) \leq H(y, \theta|x). \tag{4}$$

Intuitively, the lower bound can be justified by the fact that conditioning on more variables can only decrease or keep entropy the same. The upper bound follows from Equation 3, and $H(\theta|x, y) \geq 0$ since $\theta$ is modeled as a discrete random variable (ensemble index), unlike $x$ and $y$ which are continuous.

### 3.2 PaiDEs Definition

Let $D_\rho(p_i \parallel p_j)$ denote a generalized distance function between the probability distributions $p_i$ and $p_j$, which for our case represent $p_i = p(y|x, \theta_i)$ and $p_j = p(y|x, \theta_j)$, respectively. More specifically, $D$ is referred to as a premetric, $D_\rho(p_i \parallel p_j) \geq 0$ and $D_\rho(p_i \parallel p_j) = 0$ if $p_i = p_j$. The distance function need not be symmetric nor obey the triangle inequality. As such, PaiDEs can be defined as,

$$\hat{H}_\rho(y|x) := H(y|\theta, x) - \sum_{i=1}^{M} \pi_i \ln \sum_{j=1}^{M} \pi_j \exp\left(-D_\rho(p_i \parallel p_j)\right). \tag{5}$$

PaiDEs have many options for $D_\rho(p_i \parallel p_j)$ (Kullback-Leibler divergence, Wasserstein distance, Bhattacharyya distance, Chernoff $\alpha$-divergence, Hellinger distance, etc.).

**Theorem 3.1.** *As from Kolchinsky and Tracey [35], using the extreme distance functions,*

$$D_{min}(p_i \parallel p_j) = 0 \quad \forall i, j$$

$$D_{max}(p_i \parallel p_j) = \begin{cases} 0, & if\, p_i = p_j, \\ \infty, & otherwise, \end{cases}$$

*one can show that PaiDEs lie within bounds for entropy established in Equation 4.*

Refer to Appendix A for the proof of Theorem 3.1. This provides a general class of estimators but a distance function still needs to be chosen.

### 3.3    Tighter Bounds for PaiDEs

Let the Chernoff $\alpha$-divergence, where $\alpha \in [0, 1]$, be defined as [44],

$$D_{C_\alpha}(p_i \parallel p_j) = -\ln \int p^\alpha(y|x, \theta_i) p^{1-\alpha}(y|x, \theta_j) dy.$$

**Corollary 3.2.** *As from Kolchinsky and Tracey [35], when applying Chernoff $\alpha$-divergence as our distance function in Equation 5, we achieve a tighter lower bound than $H(y|\theta, x)$ from Equation 4,*

$$H(y|\theta, x) \leq \hat{H}_{C_\alpha}(y|x) \leq H(y|x). \tag{6}$$

Refer to Appendix A for the proof of Corollary 3.2. In addition, the tightest lower bound can be shown to be $\alpha = 0.5$ for certain situations [35]. This is known as the Bhattacharyya distance,

$$D_B(p_i||p_j) = -\ln \int \sqrt{p(y|x, \theta_i) p(y|x, \theta_j)} dy. \tag{7}$$

In addition to the improved lower bound, there is an improved upper bound as well. Let Kullback-Liebler (KL) divergence be defined as follows,

$$D_{KL}(p_i \parallel p_j) = \int p(y|x, \theta_i) \ln \frac{p(y|x, \theta_i)}{p(y|x, \theta_j)} dy.$$

Note that the KL divergence is a not metric but does suffice as a generalized distance function.

**Corollary 3.3.** *As from Kolchinsky and Tracey [35], when applying Kullback-Liebler divergence as our distance function in Equation 5, we achieve a tighter upper bound than $H(y, \theta|x)$ from Equation 4,*

$$H(y|x) \leq \hat{H}_{KL}(y|x) \leq H(y, \theta|x). \tag{8}$$

Refer to Appendix A for the proof of Corollary 3.3.

## 4    Estimating Epistemic Uncertainty with Pairwise Epistemic Estimators

Our extension of prior methodologies enables the estimation of epistemic uncertainty, without the need for sampling. We illustrate this capability for NFs in the main paper and for Probabilistic Network Ensembles (PNEs) in Appendix E [39]. Note that our proposed estimators are applicable to any ensemble model whose component output distributions have closed-form pairwise distances. Since many models have these properties, we focus on Nflows Base for its expressiveness with multimodal aleatoric uncertainty and on widely used PNEs.

### 4.1    Pairwise Epistemic Estimators

By applying our definition of PaiDEs to Equation 1, we obtain the following expression:

$$\hat{I}_\rho(y, \theta) = \hat{H}_\rho(y|x) - H(y|x, \theta) = -\sum_{i=1}^{M} \pi_i \ln \sum_{j=1}^{M} \pi_j \exp\left(-D_\rho(p_i \parallel p_j)\right). \tag{9}$$

PaiDEs estimate epistemic uncertainty using only pairwise component distances, removing reliance on sampling. We propose two estimators:

$$\hat{I}_B(y,\theta) = -\sum_{i=1}^{M} \pi_i \ln \sum_{j=1}^{M} \pi_j \exp\left(-D_B(p_i \parallel p_j)\right),$$

$$\hat{I}_{KL}(y,\theta) = -\sum_{i=1}^{M} \pi_i \ln \sum_{j=1}^{M} \pi_j \exp\left(-D_{KL}(p_i \parallel p_j)\right),$$

where $D_B(p_i \parallel p_j)$ and $D_{KL}(p_i \parallel p_j)$ are defined for Gaussians in Appendix B. We refer to $\hat{I}_B(y,\theta)$ as PairEpEst-Bhatt and $\hat{I}_{KL}(y,\theta)$ as PairEpEst-KL.

## 4.2 Nflows Base

In this study, we utilize an ensemble technique known as Nflows Base, which has previously demonstrated robust performance in estimating both aleatoric and epistemic uncertainty on simulated robotic datasets by leveraging NFs to create ensembles [4].

NFs have traditionally been applied to unsupervised tasks [54, 53]. However, NFs have also been adapted for supervised learning tasks, particularly for regression [59, 1]. Using the structure described in Winkler et al. [59], a supervised NF is defined as:

$$p_{y|x}(y|x) = p_{b|x,\theta}(g_\phi^{-1}(y,x))|\det(J(g_\phi^{-1}(y,x)))|,$$

$$\log(p_{y|x}(y|x)) = \log(p_{b|x,\theta}(g_\phi^{-1}(y,x))) + \log(|\det(J(g_\phi^{-1}(y,x)))|),$$

where $p_{y|x}$ is the output distribution, $p_{b|x,\theta}$ is the base distribution with parameters $\theta$, $J$ refers to the Jacobian, and $g_\phi^{-1} : y \times x \mapsto b$ is the bijective mapping with parameters $\phi$. For a complete review of NFs, refer to Papamakarios et al. [47]. Nflows Base creates an ensemble in the base distribution,

$$p_{y|x,\theta}(y|x,\theta_j) = p_{b|x,\theta_j}(g_\phi^{-1}(y,x))|\det(J(g_\phi^{-1}(y,x)))|,$$

where $p_{b|x,\theta_j}(b|x,\theta_j) = N(\mu_{\theta_j}(x), \Sigma_{\theta_j}(x))$, $\mu_{\theta_j}(x)$ and $\Sigma_{\theta_j}(x)$ denote the mean and covariance for input $x$ and conditioned on $\theta_j$. To encourage diversity in the ensemble, each member is initialized with different random weights, trained on a bootstrapped subset of the data, and assigned a fixed dropout mask sampled at the start of training (with $p = 0.5$), similar to [1]. This mask remains constant throughout training and inference. By constructing the ensemble within the base distribution, one can make use of closed-form pairwise-distance formulae as Berry and Meger [4] showed that estimating epistemic uncertainty in the base distribution is equivalent to estimating it in the output distribution.

Exploiting these closed-form pairwise distance formulae, Berry and Meger [4] showed that Nflows Base outperforms naive NF ensemble methods when estimating epistemic uncertainty. The aleatoric uncertainty from Equation 1 can be estimated in the base distribution space, and therefore can be computed analytically. However, this approach does not extend to the total uncertainty in Equation 1, which is why MC techniques have traditionally been used to estimate epistemic uncertainty.

## 4.3 Integrating Pairwise Epistemic Estimators with Nflows Base Ensembles

By combining Nflows Base and PairEpEsts, we construct an expressive non-parametric model capable of capturing intricate aleatoric uncertainty in the output distribution while efficiently estimating epistemic uncertainty in the base distribution. Equation 9 then becomes,

$$\hat{I}_\rho(y,\theta) = -\sum_{i=1}^{M} \pi_i \ln \sum_{j=1}^{M} \pi_j \exp\left(-D_\rho(p_{b|x,\theta_j} \parallel p_{b|x,\theta_i})\right). \tag{10}$$

Unlike previously proposed methods, we are able to estimate epistemic uncertainty without taking a single sample. Figure 7 in the Appendix shows an example of the distributional pairs that need to be considered in order to estimate epistemic uncertainty for an Nflows Base model. By applying PairEpEsts in the base distribution space we are able to capture epistemic uncertainty in the output space [4]. This approach enables the use of established formulae for computing distributional distances.

PairEpEsts can in principle be extended to skewed or heavy-tailed base distributions, provided that the pairwise premetric admits a closed-form expression. Recent work on tail-adaptive normalizing flows [40, 25] offers a promising direction for incorporating heavy-tailed behavior into normalizing flows. Integrating these approaches with PairEpEsts represents an important avenue for future research, particularly in domains where tail risk is critical.

## 4.4 Integrating Pairwise Epistemic Estimators with Probabilistic Network Ensembles

In addition to Nflows Base ensembles, we extend PairEpEsts to PNEs, which are commonly used for uncertainty quantification in regression [12, 38, 39]. PNEs comprise multiple components that produce Gaussian predictive distributions with learned means and variances.

Applying PairEpEsts to PNEs leverages the closed-form pairwise distances between these Gaussian outputs to estimate epistemic uncertainty without sampling. Formally, the estimator is:

$$\hat{I}_\rho(y, \theta) = -\sum_{i=1}^{M} \pi_i \ln \sum_{j=1}^{M} \pi_j \exp\big(-D_\rho(p_{y|x,\theta_j} \parallel p_{y|x,\theta_i})\big),$$

where $p_{y|x,\theta_i}$ denotes the $i$-th output distribution. Our experiments in Appendix E show that PairEpEsts with PNEs achieve comparable or better epistemic uncertainty estimates than base-lines, especially in high-dimensional settings. Although PNEs are less expressive than normalizing flow ensembles, their combination with PairEpEsts offers a practical and efficient framework for scalable uncertainty quantification across diverse ensemble models.

# 5 Experimental Results

To evaluate our method, we tested each of our PairEpEsts (KL and Bhatt) on two 1D environments, as has been previously proposed in the literature [15]. Additionally, we present 4 multi-dimensional environments. In contrast to previous papers [48, 2], we evaluate each method on high dimensional output space (up to 257) to demonstrate the utility of PairEpEsts in this setting. Note that, for all experiments, the model components are assumed to be uniform, $\pi_j = \frac{1}{M}$, independent of $x$. All model hyper-parameters are contained in Appendix B and the code can be found in the supplementary material. The experimental design follow previous literature [4].

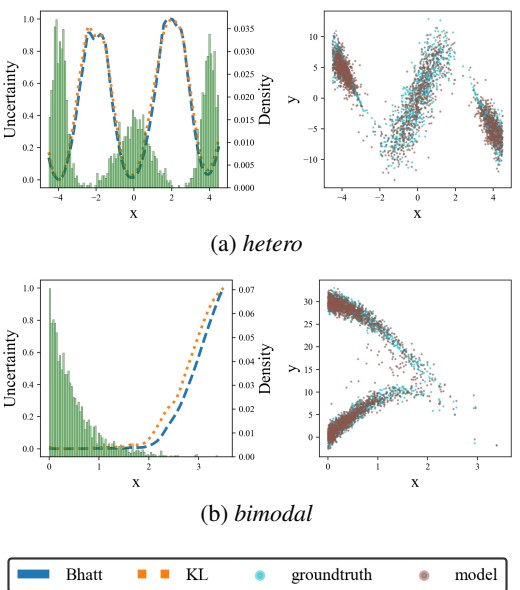

(a) *hetero*

(b) *bimodal*

Figure 2: In the right graphs, the brown dots are sampled from Nflows Base and the cyan dots are the ground-truth data. The left graphs depict the epistemic uncertainty estimates corresponding to our two proposed estimators and the density of the ground-truth data in the green histogram.

## 5.1 Data

We evaluated our estimators on two 1D benchmarks, *hetero* and *bimodal*. The ground-truth data for *hetero* and *bimodal* can be seen in Figure 2 on the right graphs with the cyan dots. For *hetero*, there are two regions with low density (2 and -2) as can be seen by the green bar chart in the left graph which corresponds to the density of the ground-truth data. In these regions, one would expect a model to have high epistemic uncertainty. For *bimodal*, the number of data points drops off as x increases, thus we would expect a model to have epistemic uncertainty grow as x does. All details for data generation are contained in Appendix C.

In addition to the 1D environments, we tested our methods on four multi-dimensional environments: *Pendulum*, *Hopper*, *Ant*, and *Humanoid* [55]. Replay buffers were collected and the dynamics for each environment was modeled, $f_\theta(s_t, a_t) = \hat{s}_{t+1}$. The choice of multi-dimensional environments is

Table 1: Mean RMSE on the test set for the last ($100^{th}$) Acquisition Batch for Nflows Base. Experiments were across ten different seeds and the results are expressed as mean $\pm$ standard deviation. The best means are in bold and results that are statistically significant are highlighted.

|  | *hetero* | *bimodal* | *Pendulum* | *Hopper* | *Ant* | *Humanoid* |
|---|---|---|---|---|---|---|
| Output Dim. | 1 | 1 | 3 | 11 | 32 | 257 |
| Random | $1.56 \pm 0.14$ | $6.4 \pm 0.62$ | $0.15 \pm 0.04$ | $0.97 \pm 0.2$ | $1.05 \pm 0.1$ | $6.59 \pm 1.54$ |
| BatchBALD | $1.54 \pm 0.16$ | $6.42 \pm 0.65$ | $0.13 \pm 0.05$ | $0.87 \pm 0.27$ | $0.94 \pm 0.03$ | $5.33 \pm 1.2$ |
| BADGE | $\mathbf{1.44} \pm 0.12$ | $6.01 \pm 0.04$ | $0.34 \pm 0.15$ | $1.11 \pm 0.27$ | $0.91 \pm 0.04$ | $7.31 \pm 3.11$ |
| BAIT | $1.51 \pm 0.14$ | $6.26 \pm 0.33$ | $0.17 \pm 0.05$ | $1.06 \pm 0.5$ | $0.94 \pm 0.04$ | $11.01 \pm 0.23$ |
| MC (BALD) | $1.54 \pm 0.17$ | $6.01 \pm 0.04$ | $0.06 \pm 0.01$ | $0.32 \pm 0.02$ | $1.01 \pm 0.05$ | $10.71 \pm 0.43$ |
| KL (ours) | $1.47 \pm 0.15$ | $6.01 \pm 0.04$ | $\mathbf{0.05} \pm 0.04$ | $0.3 \pm 0.03$ | $\mathbf{0.9} \pm 0.07$ | $3.4 \pm 0.48$ |
| Bhatt (ours) | $1.55 \pm 0.31$ | $\mathbf{6.0} \pm 0.04$ | $\mathbf{0.05} \pm 0.01$ | $\mathbf{0.29} \pm 0.03$ | $0.93 \pm 0.08$ | $\mathbf{3.37} \pm 0.44$ |

⬤ $p < 0.05$  ⬤ $p < 0.01$  ⬤ $p < 0.001$

motivated by their common use as benchmarks and their higher-dimensional output space, providing a robust validation of our methods. Note that, for *Ant* and *Humanoid*, the dimensions representing their contact forces were eliminated due to a bug in Mujoco-v2.

In the Mujoco environment, the input consists of the current state and action, which are used to predict the next state based on the dynamics of the environment. In the hetero- and bimodal environments, we simplify the task to take a single input variable x, and predict the output variable y.

## 5.2 1D Experiments

Our 1D environments provide empirical proof that PairEpEsts can accurately measure epistemic uncertainty. Figure 2 depicts that both PairEpEsts are proficient at estimating the epistemic uncertainty as each method shows an increase in epistemic uncertainty around 2 and -2 on the *hetero* setting. This can be seen from the blue and orange lines with both estimators performing similarly.

A similar pattern can be seen for the *bimodal* setting in Figure 2, which shows that both PairEpEsts can accurately capture epistemic uncertainty. Each estimator shows the pattern of increasing epistemic uncertainty where the data is more scarce. Both examples show accurate epistemic uncertainty estimation with no loss in aleatoric uncertainty representation, as demonstrated in the right graphs in Figure 2: the brown dots closely match the cyan dots. Note that for visual clarity, the epistemic uncertainty has been scaled to 0-1, while ensuring that the relative properties of the estimators are preserved. Further discussion on over- and underestimation is provided in Section 6. While simple count-based heuristics may suffice in toy 1D settings like Figure 2, they fail to scale to higher-dimensional spaces where data is sparse and frequency statistics are uninformative. Moreover, they cannot exploit learned representations of the model, which are essential for structured, high-dimensional control tasks such as Hopper, Ant, and Humanoid. For these reasons, principled estimators such as PairEpEsts are required in realistic settings.

## 5.3 Active Learning

While the 1D experiments provide evidence of our estimators' effectiveness for estimating epistemic uncertainty, the active learning experiments extend this evaluation to higher-dimensional data and for a decision-making task. In order to benchmark our method, we compare against four state-of-the-art active learning frameworks: BatchBALD which maximizes information gain over batches using MC approximations [34], BADGE which uses gradient-based selection to identify the most informative data points for model improvement [3] and BAIT selects data based on uncertainty estimation through Bayesian active learning [2]. Additionally, a random baseline is included. The training set is initialized with 100 or 200 data points for 1D and multi-dimensional environments, respectively. In each acquisition batch, 10 points are added. For the MC estimator, $10^3$ candidate inputs were sampled to construct each acquisition batch, with their epistemic uncertainties estimated using $K = 5000$.

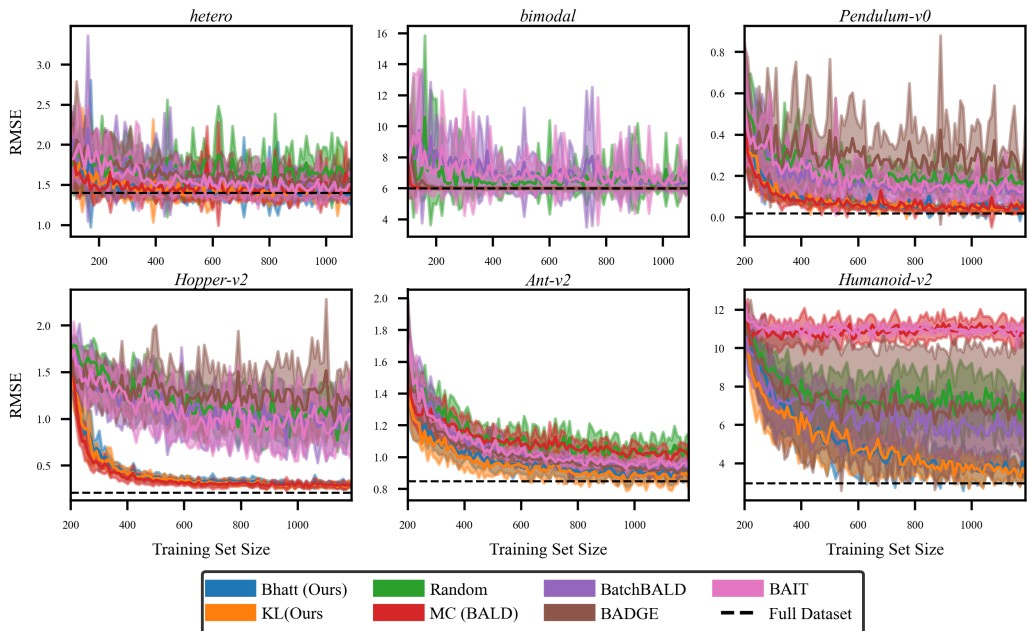

Figure 3: Mean RMSE on the test set as additional data is incorporated into the training set for Nflows Base. Both proposed methods perform comparably or significantly better than baselines, particularly in high-dimensional settings.

In the *Humanoid* environment, only 100 candidates were used due to computational constraints. In contrast, PairEpEsts sampled $10^4$ candidate inputs and estimated their epistemic uncertainties for each environment including *Humanoid*. This underscores one advantage of our estimators over MC estimators, as they can estimate epistemic uncertainty over larger regions at a lower computational cost than their MC counterparts. Note that the other benchmarks each sampled $10^4$ inputs as well and their respective acquisition functions applied. The RMSE on the test set was calculated at each acquisition batch.

Table 1 shows the performance of each framework on the 100th acquisition batch. Welch's t-test, with a Holm–Bonferroni correction (see Appendix G), compares our estimators to baselines in each environment. PairEpEsts achieve lower or comparable RMSEs, demonstrating their efficacy in estimating epistemic uncertainty. In higher dimensions, our estimators significantly outperform other methods, as shown in Humanoid. Learning curves are presented in Figure 3. Note that in the *bimodal* setting, all methods converge to similar performance and are indistinguishable in the plot. BatchBALD and BADGE, originally designed for classification, required adaptation for regression, facing challenges due to differences in uncertainty measure computation. Similarly, BAIT, designed for 1D regression, did not perform well in high-dimensional settings. The dashed line indicates the performance of our model when trained on the full dataset, serving as an upper bound for comparison.

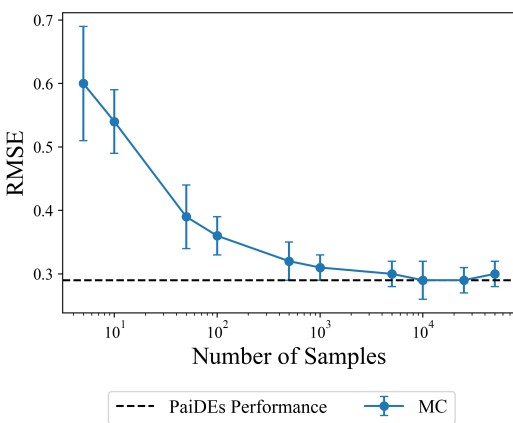

Figure 4: RMSE on the test set at the $100^{th}$ acquisition batch of the MC estimator on the *Hopper* environment for Nflows Base as the number of samples, $K$, per input, $x$, increases. Experiment were run across 10 seeds and the mean and stdev is being reported.

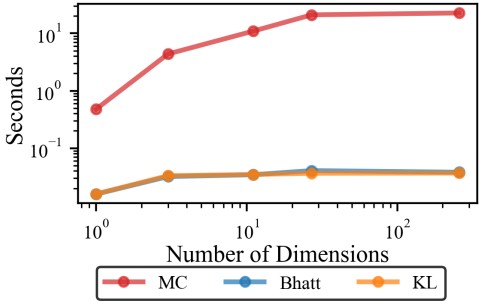

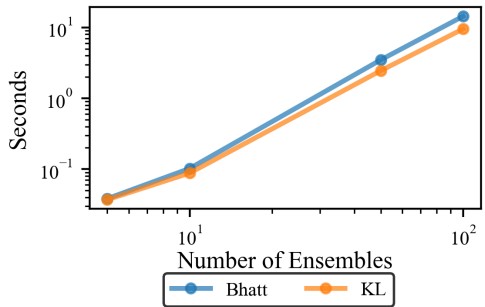

Figure 5: The amount of time taken for Nflows Base for each estimator across the different settings (1, 3, 11, 27, 257 dimensions). Results are averaged over 10 seeds. MC results use $K = 5000$ samples per acquisition candidate across all environments.

Figure 6: The amount of time taken for PaiDEs for Nflows Base as the number of ensemble components increases for *Humanoid-v2*. Results are averaged over 10 seeds.

For a deeper analysis of our method, we compared our estimators to MC estimators with a varying sample size, $K$, in the *Hopper* environment. As expected, MC estimators perform on par with PairEpEsts with a sufficient number of samples. This trend is illustrated in Figure 4. The MC estimator reaches the peak performance of our estimators given enough samples but does not perform better than PairEpEsts, suggesting that our estimators are sufficient to estimate epistemic uncertainty.

## 5.4 Time Analysis

In addition to benchmarking our estimators on active learning experiments, we provide an analysis of the time gains across our experiments. Figure 5 depicts the speed increase that can be gained using PairEpEsts over an MC approach. PairEpEsts provide a 1–2 order of magnitude runtime improvement over Monte Carlo approaches. The estimates are obtained from the active learning experiments.

## 6 Limitations

PairEpEsts have quadratic computational cost as the number of components grows. For asymmetric distances like KL-divergence, $M^2 - M$ pairwise distances are computed, while symmetric distances like Bhattacharyya require only $\frac{M^2 - M}{2}$. Figure 6 shows timing as ensemble size increases. Bhattacharyya's cost could be further reduced using symmetry. Despite this, deep learning ensembles usually have few components (5–10) [45, 12], making the complexity manageable. The limitation is more relevant in large ensembles, such as weather forecasting [57].

PairEpEsts introduce a small bias absent in MC estimators. However, since active learning depends on the relative ordering of acquisition scores, this bias does not affect performance. As shown in Appendix D, the rankings are well preserved (Table 5). The relationships are illustrated in Figure 8, with epistemic uncertainty values normalized between 0 and 1 in Figure 2.

## 7 Related Work

Bayesian neural networks with information-based criteria are widely used for active learning in image classification [21, 32, 34], while others use gradient embeddings [3, 2]. Most focus on classification, with few methods extending to 1D regression [2]. Our work tackles active learning in high-dimensional regression, filling a gap relevant to robotic locomotion. Despite limitations of mutual information in classification [58], we demonstrate its value in high-dimensional regression.

Ensembles have been harnessed for epistemic uncertainty estimation [39, 11, 12]. Specifically related to our work, ensembles have been leveraged to quantify epistemic uncertainty in regression problems and active learning [15, 48, 4]. Depeweg et al. [15] employed Bayesian neural networks to model mixtures of Gaussians and demonstrated their ability to measure uncertainty in low-dimensional environments (1-2D). Building upon this foundation, Postels et al. [48] and Berry and Meger [4]

extended the research by developing efficient NF ensemble models that capture epistemic uncertainty. Our work advances this line of research by eliminating the need for sampling to estimate epistemic uncertainty, resulting in a faster and more effective method, especially in higher dimensions.

In addition to Bayesian neural networks and ensemble methods, the concept of Credal Bayesian Deep Learning has been introduced to enhance uncertainty quantification [7]. Furthermore, distributionally robust statistical verification has been proposed for high-dimensional autonomous systems using Imprecise Neural Networks, which provide uncertainty quantification guarantees [19].

Entropy estimators, which do not rely on sampling, is an active area of research [30, 31, 28, 35]. Kulak et al. [37] and Kulak and Calinon [36] demonstrated the utility of PaiDEs within Bayesian contexts, employing PaiDEs to estimate conditional predictive posterior entropy. In contrast, our approach provides a more general estimate of epistemic uncertainty, as defined in Equation 1, which can be applied to both ensemble, Bayesian methods and deep learning models.

Several methods have emerged in the literature for estimating epistemic uncertainty without relying on sampling techniques [56, 8]. Both Van Amersfoort et al. [56] and Charpentier et al. [8] focus on classification tasks with 1D categorical outputs. Charpentier et al. [9] extends the work of Charpentier et al. [8] to regression tasks but is limited to modeling outputs as members of the exponential family. In contrast, our approach can handle more complex output distributions by directly considering the outputs from NFs and can be applied to a larger space of regression models.

## 8 Conclusions

In this study, we introduced epistemic uncertainty estimators and applied them to Active Learning. We depicted how our method can be used to more efficiently quantify uncertainty by leveraging closed-form formulae instead of sampling. This led to improvements in computational speed and accuracy. As learning becomes pervasive in high-dimensional tasks in society, our method is well-placed to enable epistemic uncertainty awareness without unnecessary compute.

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

# A Proofs

The proofs follow from the steps of Kolchinsky and Tracey [35].

**Proof of Theorem 3.1**

*Proof.* Applying $D_{min}(p_i \parallel p_j)$ as our distance in PaiDEs,

$$\hat{H}(Y|X) = H(Y|X,\Theta) - \sum_{i=1}^{M} \pi_i \ln \sum_{j=1}^{M} \pi_j \exp\left(-D_{min}(p_i \parallel p_j)\right)$$

$$= H(Y|X,\Theta) - \sum_{i=1}^{M} \pi_i \ln \sum_{j=1}^{M} \pi_j$$

$$= H(Y|X,\Theta).$$

Note the last line follows from the fact that the component weights must sum to one, $\sum_{j=1}^{M} \pi_j = 1$. Next using $D_{max}(p_i \parallel p_j)$,

$$\hat{H}(Y|X) = H(Y|X,\Theta) - \sum_{i=1}^{M} \pi_i \ln \sum_{j=1}^{M} \pi_j \exp\left(-D_{max}(p_i \parallel p_j)\right)$$

$$= H(Y|X,\Theta) - \sum_{i=1}^{M} \pi_i \ln \left(\pi_i + \sum_{j\neq i} \pi_j \exp\left(-D_{max}(p_i \parallel p_j)\right)\right)$$

$$= H(Y|X,\Theta) - \sum_{i=1}^{M} \pi_i \ln \pi_i$$

$$= H(Y|X,\Theta) + H(\Theta|X)$$

$$= H(Y,\Theta|X).$$

This shows the upper bound with $D_{max}(p_i \parallel p_j)$ and the lower bound with $D_{min}(p_i \parallel p_j)$. Note that in our case $H(\Theta|X) = H(\Theta)$ as the distribution of weights does not depend on our input. □

**Proof of Corollary 3.2**

*Proof.* To show the upper bound from Equation 6 we use a derivation from Haussler and Opper [23],

$$H(Y|X) = H(Y|X,\Theta) - \int \sum_{i=1}^{M} \pi_i p_i(y|x) \ln \frac{p_{Y|X}(y|x)}{p_i(y|x)} dy$$

$$= H(Y|X,\Theta) - \int \sum_{i=1}^{M} \pi_i p_i(y|x) \ln \frac{p_{Y|X}(y|x)}{p_i(y|x)^{1-\alpha} \sum_j \pi_j p_j(y|x)^{1-\alpha}} dy$$

$$- \int \sum_{i=1}^{M} \pi_i p_i(y|x) \ln \frac{\sum_j \pi_j p_j(y|x)^{1-\alpha}}{p_i(y|x)^{\alpha}} dy$$

$$\geq H(Y|X,\Theta) - \int \sum_{i=1}^{M} \pi_i p_i(y|x) \ln \frac{\sum_j \pi_j p_j(y|x)^{1-\alpha}}{p_i(y|x)^{\alpha}} dy$$

$$\geq H(Y|X,\Theta) - \sum_{i=1}^{M} \pi_i \ln \int p_i(y|x)^{\alpha} \sum_j \pi_j p_j(y|x)^{1-\alpha} dy$$

$$= H(Y|X,\Theta) - \sum_{i=1}^{M} \pi_i \ln \sum_{j=1}^{M} \pi_j \exp\left(-C_\alpha(p_i \parallel p_j)\right).$$

The two inequalities follow from Jensen's inequality. □

**Proof of Corollary 3.3**

*Proof.* The lower bound can be shown using the definition of entropy for a mixture,

$$H(Y|X) = -\sum_{i=1}^{M} \pi_i E \left[ \ln \sum_{j=1}^{M} \pi_j p_j(y|x) \right]$$

$$\leq -\sum_{i=1}^{M} \pi_i \ln \sum_{j=1}^{M} \pi_j \exp\left( E_{p_i} \left[ \ln p_j(y|x) \right] \right)$$

$$= -\sum_{i=1}^{M} \pi_i \ln \sum_{j=1}^{M} \pi_j \exp\left( -H(p_i \| p_j) \right)$$

$$= \sum_{i=1}^{M} \pi_i H(p_i) - \sum_{i=1}^{M} \pi_i \ln \sum_{j=1}^{M} \pi_j \exp\left( -KL(p_i \| p_j) \right)$$

$$= H(Y|X, \Theta) - \sum_{i=1}^{M} \pi_i \ln \sum_{j=1}^{M} \pi_j \exp\left( -KL(p_i \| p_j) \right),$$

where $E_{p_i}$ refers to the expectation when $Y$ is distributed as $p_i$ and $H(p_i \| p_j)$ indicates the cross entropy. The inequality in line 2 follows from Hershey and Olsen [24] and Paisley [46]. □

## A.1 PairEpEst-KL vs EPKL

In addition to the previous results from Kolchinsky and Tracey [35], we provide a unique proof to our paper demonstrating that the PairEpEst-KL estimator provides a strictly tighter upper bound on MI than the Expected Pairwise KL (EPKL) introduced by Malinin and Gales [42]. EPKL is another uncertainty measure that captures epistemic uncertainty by averaging the pairwise Kullback-Leibler divergences between predictive distributions of ensemble members. It quantifies how diverse the ensemble's predictions are, reflecting the model's uncertainty due to lack of knowledge or data. EPKL serves as an upper bound on MI. Our PairEpEst-KL estimator leverages a log-sum-exp formulation, yielding a sharper (tighter) upper bound on MI, which leads to more precise uncertainty quantification.

Specifically, consider an ensemble of $M$ predictive distributions $p_1(y), \ldots, p_M(y)$. The EPKL is defined as the average pairwise Kullback-Leibler divergence between distinct ensemble components:

$$\text{EPKL} := \frac{1}{M(M-1)} \sum_{i \neq j} D_{\text{KL}}(p_i(y) \| p_j(y)).$$

Our PairEpEst-KL estimator, denoted by $B$, is given by

$$B := -\frac{1}{M} \sum_{i=1}^{M} \ln \left( \frac{1}{M} \sum_{j=1}^{M} \exp\left( - D_{\text{KL}}(p_i(y) \| p_j(y)) \right) \right).$$

The following theorem formalizes the relationship between these two quantities and establishes that

$$B < \text{EPKL},$$

meaning that PairEpEst-KL yields a strictly sharper upper bound on MI compared to EPKL.

**Theorem A.1.** *Let $p_1(y), \ldots, p_M(y)$ be probability distributions, and define*

$$A := \frac{1}{M(M-1)} \sum_{i \neq j} D_{\text{KL}}(p_i(y) \| p_j(y)),$$

*and*

$$B := -\frac{1}{M} \sum_{i=1}^{M} \ln \left( \frac{1}{M} \sum_{j=1}^{M} \exp\left( - D_{\text{KL}}(p_i(y) \| p_j(y)) \right) \right).$$

*Then,*

$$B \leq \frac{1}{M^2} \sum_{i,j} D_{\mathrm{KL}}(p_i(y)\|p_j(y)) < A,$$

*and in particular,*

$$B < A.$$

*Proof.* Define

$$s_{ij} := -D_{\mathrm{KL}}(p_i(y)\|p_j(y)) \leq 0.$$

By Jensen's inequality applied to the concave function $\ln(\cdot)$, for each fixed $i$,

$$\ln\left(\frac{1}{M}\sum_{j=1}^{M} e^{s_{ij}}\right) \geq \frac{1}{M}\sum_{j=1}^{M} s_{ij}.$$

Multiplying both sides by $-1$ reverses the inequality:

$$-\ln\left(\frac{1}{M}\sum_{j=1}^{M} e^{s_{ij}}\right) \leq -\frac{1}{M}\sum_{j=1}^{M} s_{ij}.$$

Substituting back for $s_{ij}$, we get

$$-\ln\left(\frac{1}{M}\sum_{j=1}^{M} \exp\big(-D_{\mathrm{KL}}(p_i(y)\|p_j(y))\big)\right) \leq \frac{1}{M}\sum_{j=1}^{M} D_{\mathrm{KL}}(p_i(y)\|p_j(y)).$$

Averaging over $i$, it follows that

$$B = \frac{1}{M}\sum_{i=1}^{M}\left[-\ln\left(\frac{1}{M}\sum_{j=1}^{M} e^{-D_{\mathrm{KL}}(p_i\|p_j)}\right)\right] \leq \frac{1}{M^2}\sum_{i=1}^{M}\sum_{j=1}^{M} D_{\mathrm{KL}}(p_i\|p_j).$$

Since $D_{\mathrm{KL}}(p_i\|p_i) = 0$, the double sum over all $i, j$ is equal to the sum over $i \neq j$:

$$\sum_{i=1}^{M}\sum_{j=1}^{M} D_{\mathrm{KL}}(p_i\|p_j) = \sum_{i \neq j} D_{\mathrm{KL}}(p_i\|p_j).$$

Note that

$$M^2 > M(M-1) \implies \frac{1}{M^2} < \frac{1}{M(M-1)},$$

so

$$B \leq \frac{1}{M^2}\sum_{i \neq j} D_{\mathrm{KL}}(p_i\|p_j) < \frac{1}{M(M-1)}\sum_{i \neq j} D_{\mathrm{KL}}(p_i\|p_j) = A,$$

which establishes the strict inequality

$$B < A,$$

completing the proof. □

## B  Compute and Hyper-parameter Details

The Nflows Base model employed one nonlinear transformation, $g$, with a single hidden layer containing 20 units, utilizing cubic spline flows as per [17]. The base network consisted of two hidden layers, each comprising 40 units with ReLU activation functions. It is important to note that all base distributions were Gaussian. The PNEs adopted an architecture of three hidden layers each with 50 units and ReLU activation functions. Model hyperparameters remained consistent across all experiments. The base distribution and the bijective mapping are trained simultaneously. Each base distribution network is mapped through the same bijective mapping, which is shared across all ensemble components. Every time a base distribution network takes a training step, the bijective mapping is updated accordingly. This approach ensures that the bijective mapping

**Algorithm 1** Active Learning Using PairEpEsts

---

**Input:** training dataset $\mathcal{D}_{train}$, oracle dataset $\mathcal{D}_{oracle}$, number of iterations $T$, sample size $S$, and batch size $B$
**for** i=1,2,...,T **do**
    Randomly initialize each ensemble component $\theta_j$.
    Train our ensemble on $\mathcal{D}_{train}$ using bootstrapped samples.
    Sample $S$ points from $\mathcal{D}_{oracle}$ uniformly, $\mathcal{D}_S$.
    Calculate the top $B$ values according to Equation 9 from $\mathcal{D}_S$, $\mathcal{D}_B$.
    Add $\mathcal{D}_B$ to the training dataset while removing it from the oracle dataset, $\mathcal{D}_{train} = \mathcal{D}_B \cup \mathcal{D}_{train}$
    and $\mathcal{D}_{oracle} = \mathcal{D}_{oracle} \setminus \mathcal{D}_B$
**end for**
train final model on $\mathcal{D}_{train}$.

---

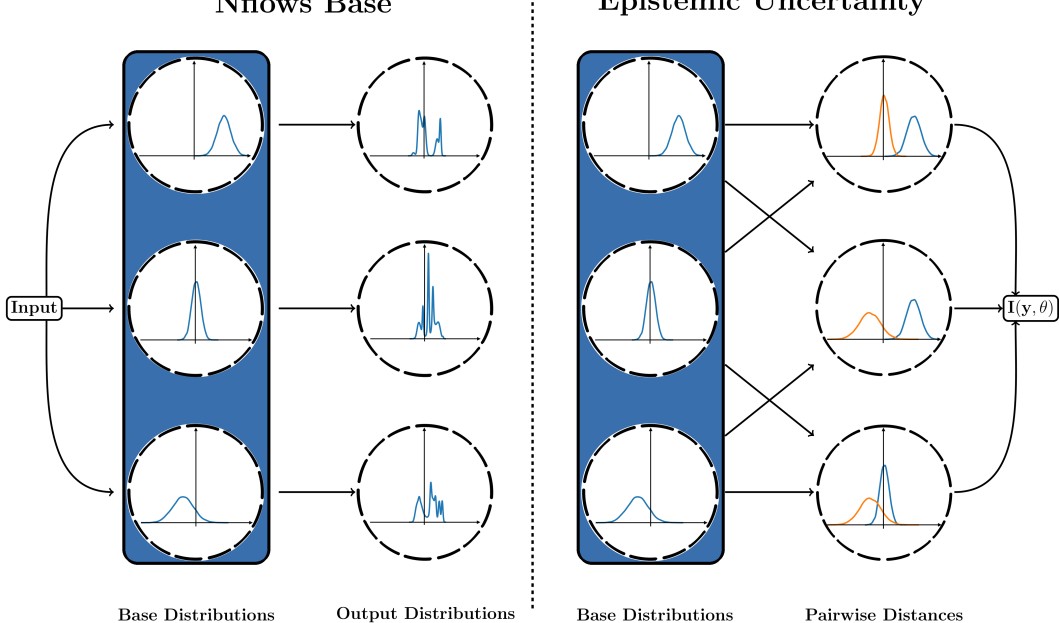

Figure 7: Nflows Base as an ensemble of 3 components with one bijective transformation on the left and an example of the pairwise comparisons needed to estimate epistemic uncertainty for said model on the right. Note the base distributions, instead of the output distributions, from Nflows Base are used to estimate the epistemic uncertainty which are highlighted by the blue bar.

is consistent across all components of the ensemble. This methodology is similar to previous approaches, such as those proposed by Osband et al. [45], where shared mappings are utilized within ensemble-based models. Training was conducted using 16GB RAM on Intel Gold 6148 Skylake @ 2.4 GHz CPUs and NVidia V100SXM2 (16G memory) GPUs. For each experimental setting, PNEs and Nflows Base were executed with five ensemble components. The MC estimator sampled 1000 and 5000 points for Nflows Base and PNEs, respectively, for each $x$ conditioned on. The nflows library [18] was employed with minor modifications. Our code can be found at https://github.com/nwaftp23/pairflow-uncertainty.

Note that for the Bhatt estimator, the Bhattacharyya distance between two Gaussians is,

$$D_B(p_i||p_j) = \frac{1}{8}(\mu_{i|x} - \mu_{j|x})^{\mathrm{T}}\Sigma^{-1}(\mu_{i|x} - \mu_{j|x}) + \frac{1}{2}\ln\left(\frac{\det\Sigma}{\sqrt{\det\Sigma_{i|x}\det\Sigma_{j|x}}}\right),$$

$$\Sigma = \frac{\Sigma_{i|x} + \Sigma_{j|x}}{2}.$$

Also note that for the KL estimator, the KL divergence between two Gaussians is,

$$D_{KL}(p_i \parallel p_j) = \frac{1}{2} \left( \text{tr}(\Sigma_{j|x}^{-1} \Sigma_{i|x}) - D + \ln \left( \frac{\det \Sigma_{j|x}}{\det \Sigma_{i|x}} \right) + (\mu_{j|x} - \mu_{i|x})^{\mathsf{T}} \Sigma_{j|x}^{-1} (\mu_{j|x} - \mu_{i|x}) \right),$$

where $\text{tr}(\cdot)$ refers to the trace of a matrix.

To complement the per-step acquisition runtimes reported in Figure 3, we also provide end-to-end active learning runtimes in Table 3 and Table 4. These results reflect the total wall-clock time required for a full active learning loop in each environment. These tables reinforce the efficiency advantage of PairEpEsts: across both architectures, KL and Bhatt estimators consistently require less total runtime than MC, particularly in higher-dimensional environments such as *Ant* and *Humanoid*. Notably, the runtime gap widens with dimensionality, reflecting the scalability of PairEpEsts relative to sampling-based methods.

To justify our choice of five ensemble components, we conducted an additional experiment analyzing the effect of ensemble size on active learning performance. This experiment was performed on the *Hopper* environment, with performance measured as RMSE at the final acquisition batch. Results are reported in Table 2. Performance improves markedly when increasing from 3 to 5 members, but remains stable for larger ensembles, suggesting that five components provide a strong balance between efficiency and accuracy.

Table 2: RMSE at the final acquisition batch for different ensemble sizes on *Hopper*.

|       | 3    | 4    | 5    | 7    | 10   |
|-------|------|------|------|------|------|
| KL    | 0.51 | 0.43 | 0.30 | 0.31 | 0.29 |
| Bhatt | 0.50 | 0.45 | 0.29 | 0.28 | 0.30 |

Table 3: Total active learning runtimes (minutes) for Nflows Base across environments.

|       | Hetero | Bimodal | Pendulum | Hopper | Ant    | Humanoid |
|-------|--------|---------|----------|--------|--------|----------|
| KL    | 58.14  | 58.22   | 76.02    | 77.06  | 80.21  | 96.28    |
| Bhatt | 57.72  | 58.04   | 76.86    | 76.94  | 78.54  | 95.58    |
| MC    | 61.87  | 61.82   | 94.44    | 98.66  | 117.40 | 155.34   |

Table 4: Total active learning runtimes (minutes) for PNEs across environments.

|       | Hetero | Bimodal | Pendulum | Hopper | Ant   | Humanoid |
|-------|--------|---------|----------|--------|-------|----------|
| KL    | 22.68  | 22.50   | 28.44    | 28.60  | 30.58 | 35.24    |
| Bhatt | 22.50  | 22.36   | 28.24    | 28.88  | 30.14 | 36.04    |
| MC    | 24.42  | 24.62   | 35.22    | 46.42  | 66.96 | 84.30    |

## C Data

The *hetero* dataset was generated using a two step process. Firstly, a categorical distribution with three values was sampled, where $p_i = \frac{1}{3}$. Secondly, $x$ was drawn from one of three different Gaussian distributions ($N(-4, \frac{2}{5})$, $N(0, \frac{9}{10})$, $N(4, \frac{2}{5})$) based on the value of the categorical distribution. The corresponding $y$ was then generated as follows:

$$y = 7\sin(x) + 3z \left| \cos\left(\frac{x}{2}\right) \right|.$$

On the other hand, the *bimodal* dataset was created by sampling $x$ from an exponential distribution with parameter $\lambda = 2$, and then sampling $n$ from a Bernoulli distribution with $p = 0.5$. Based on the value of $n$, the $y$ value was determined as:

$$y = \begin{cases} 10\sin(x) + z & n = 0 \\ 10\cos(x) + z + 20 - x & n = 1 \end{cases}.$$

Note that for both *bimodal* and *hetero* data $z \sim N(0, 1)$.

Regarding the multi-dimensional environments, namely *Pendulum*, *Hopper*, *Ant*, and *Humanoid*, the training sets and test sets were collected using different approaches. The training sets were obtained by applying a random policy, while the test sets were generated using an expert policy. This methodology was employed to ensure diversity between the training and test datasets. Distribution shift is an inherent challenge in robotic dynamics, where data generated from different policies can vary significantly. These shifts, such as those observed in sim2real and imitation learning contexts. Notably, the OpenAI Gym library was utilized, with minor modifications [6].

# D  Additional Results

In addition to Table 1, we present Figure 3 showing the full active learning curves and Table 7 detailing additional acquisition batches. The trend of PairEpEsts outperforming or performing comparably to baselines is consistent across environments and acquisition batches. We also evaluated log-likelihood, a proper scoring rule [22], with results shown in Figure 9. Using 10 seeds, we report means and standard deviations. The PairEpEst-KL and PairEpEst-Bhatt estimators perform similarly or better than the MC estimator on all environments.

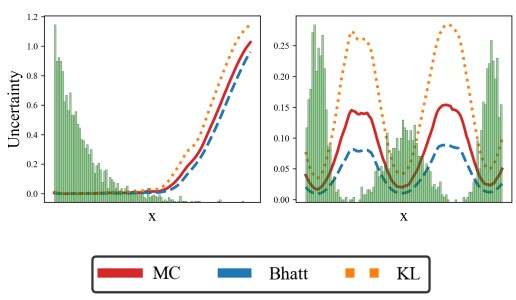

Figure 8: The bias introduced by PairEpEsts for Nflows Base compared to the MC method on the 1D environments.

For the high-dimensional *Humanoid* setting, however, log-likelihood did not improve as data was added. This limitation arises because computing ensemble log-likelihood requires evaluating each component's likelihood and summing them, a process prone to numerical underflow as values round to zero. As in Table 1, selected acquisition batches are reported in Table 8. We further compare the MC estimator and PairEpEsts on *Hopper* as the number of drawn samples increases (Figure 10). Additionally, we assessed the consistency of active learning rankings by computing Spearman's rank correlation between the MC estimator and PairEpEsts. As shown in Table 5, correlations are close to 1 across all models and datasets, indicating strong preservation of relative ordering despite small biases. All correlations are statistically significant with extremely low p-values (maximum $p = 1.11 \times 10^{-83}$).

As summarized in Table 6, Monte Carlo methods achieve comparable runtime to PairEpEsts with roughly 10 samples in *Hopper*, and with even fewer samples in higher-dimensional environments such as *Ant* and *Humanoid*. This suggests that small-sample MC can, in principle, be competitive in terms of computational cost. However, PairEpEsts provide robust and consistent uncertainty estimates without the need for sample tuning, making them particularly advantageous in high-dimensional tasks where both runtime efficiency and estimator stability are critical. Moreover, the accuracy of MC estimates deteriorates as the number of samples decreases, whereas PairEpEsts maintain reliability at low computational cost.

Table 5: Spearman's rank correlation coefficients comparing active learning rankings between MC and PairEpEsts, demonstrating that our estimators preserve relative ranking of points.

| Model | Dataset | KL Corr. | Bhatt Corr. |
|---|---|---|---|
| NFlows Base | Bimodal | 0.9958 | 0.9958 |
| NFlows Base | Hetero | 0.9976 | 0.9986 |
| PNEs | Bimodal | 0.9893 | 0.9893 |
| PNEs | Hetero | 0.9943 | 0.9972 |

Table 6: Number of MC samples that matched the runtime of PairEpEsts across environments.

| Environment | Dimensionality | MC Samples (Runtime Matched) |
|---|---|---|
| Hetero | 1 | 500 |
| Bimodal | 1 | 500 |
| Pendulum | 3 | 100 |
| Hopper | 11 | 10 |
| Ant | 27 | 5 |
| Humanoid | 257 | 5 |

Table 7: Mean RMSE on the test set for certain Acquisition Batches for Nflows Base. Experiments were across ten different seeds and the results are expressed as mean plus minus one standard deviation.

| Env | Acquisition Batch | Random | BatchBALD | BADGE | BAIT | MC (BALD) | KL (ours) | Bhatt (ours) |
|---|---|---|---|---|---|---|---|---|
| *hetero* | 10 | $1.76 \pm 0.17$ | $1.86 \pm 0.35$ | $1.84 \pm 0.36$ | $1.92 \pm 0.28$ | $\mathbf{1.48} \pm 0.2$ | $1.66 \pm 0.2$ | $1.56 \pm 0.26$ |
| | 25 | $1.69 \pm 0.45$ | $1.66 \pm 0.26$ | $1.74 \pm 0.25$ | $1.71 \pm 0.53$ | $1.44 \pm 0.1$ | $\mathbf{1.42} \pm 0.12$ | $\mathbf{1.42} \pm 0.11$ |
| | 50 | $1.64 \pm 0.22$ | $1.54 \pm 0.18$ | $1.55 \pm 0.1$ | $1.6 \pm 0.22$ | $1.45 \pm 0.12$ | $1.49 \pm 0.29$ | $\mathbf{1.42} \pm 0.11$ |
| | 100 | $1.56 \pm 0.14$ | $1.54 \pm 0.16$ | $\mathbf{1.44} \pm 0.12$ | $1.51 \pm 0.14$ | $1.54 \pm 0.17$ | $1.47 \pm 0.15$ | $1.55 \pm 0.31$ |
| *bimodal* | 10 | $8.4 \pm 3.38$ | $7.37 \pm 1.38$ | $6.11 \pm 0.1$ | $6.91 \pm 1.45$ | $6.02 \pm 0.05$ | $\mathbf{6.01} \pm 0.05$ | $\mathbf{6.01} \pm 0.05$ |
| | 25 | $6.1 \pm 0.08$ | $6.74 \pm 0.66$ | $6.02 \pm 0.04$ | $6.85 \pm 1.46$ | $6.04 \pm 0.04$ | $6.02 \pm 0.05$ | $\mathbf{6.01} \pm 0.04$ |
| | 50 | $6.57 \pm 0.72$ | $6.42 \pm 0.66$ | $6.01 \pm 0.04$ | $6.61 \pm 0.97$ | $6.01 \pm 0.04$ | $6.01 \pm 0.04$ | $\mathbf{6.0} \pm 0.04$ |
| | 100 | $6.4 \pm 0.62$ | $6.42 \pm 0.65$ | $6.01 \pm 0.04$ | $6.26 \pm 0.33$ | $6.01 \pm 0.04$ | $6.01 \pm 0.04$ | $\mathbf{6.0} \pm 0.04$ |
| *Pendulum* | 10 | $0.28 \pm 0.06$ | $0.32 \pm 0.23$ | $0.32 \pm 0.27$ | $0.42 \pm 0.19$ | $\mathbf{0.12} \pm 0.04$ | $0.15 \pm 0.06$ | $0.17 \pm 0.08$ |
| | 25 | $0.22 \pm 0.08$ | $0.17 \pm 0.06$ | $0.31 \pm 0.19$ | $0.19 \pm 0.07$ | $\mathbf{0.09} \pm 0.03$ | $\mathbf{0.09} \pm 0.03$ | $\mathbf{0.09} \pm 0.02$ |
| | 50 | $0.18 \pm 0.08$ | $0.15 \pm 0.06$ | $0.28 \pm 0.14$ | $0.17 \pm 0.06$ | $0.06 \pm 0.03$ | $0.06 \pm 0.05$ | $\mathbf{0.05} \pm 0.01$ |
| | 100 | $0.15 \pm 0.04$ | $0.13 \pm 0.05$ | $0.34 \pm 0.15$ | $0.17 \pm 0.05$ | $\mathbf{0.04} \pm 0.01$ | $0.05 \pm 0.04$ | $0.05 \pm 0.01$ |
| *Hopper* | 10 | $1.6 \pm 0.19$ | $1.3 \pm 0.26$ | $1.45 \pm 0.28$ | $1.36 \pm 0.19$ | $\mathbf{0.55} \pm 0.09$ | $0.66 \pm 0.08$ | $0.69 \pm 0.1$ |
| | 25 | $1.24 \pm 0.26$ | $1.3 \pm 0.33$ | $1.37 \pm 0.23$ | $1.18 \pm 0.29$ | $\mathbf{0.36} \pm 0.06$ | $0.38 \pm 0.05$ | $0.39 \pm 0.06$ |
| | 50 | $1.14 \pm 0.16$ | $1.06 \pm 0.3$ | $1.26 \pm 0.23$ | $1.05 \pm 0.27$ | $\mathbf{0.31} \pm 0.04$ | $0.33 \pm 0.03$ | $0.34 \pm 0.04$ |
| | 100 | $0.97 \pm 0.2$ | $0.87 \pm 0.27$ | $1.11 \pm 0.27$ | $1.06 \pm 0.5$ | $\mathbf{0.29} \pm 0.02$ | $0.3 \pm 0.03$ | $\mathbf{0.29} \pm 0.03$ |
| *Ant* | 10 | $1.33 \pm 0.1$ | $1.31 \pm 0.11$ | $1.21 \pm 0.14$ | $1.25 \pm 0.08$ | $1.24 \pm 0.13$ | $\mathbf{1.09} \pm 0.1$ | $1.13 \pm 0.09$ |
| | 25 | $1.13 \pm 0.09$ | $1.09 \pm 0.03$ | $1.05 \pm 0.04$ | $1.08 \pm 0.04$ | $1.13 \pm 0.1$ | $\mathbf{1.0} \pm 0.07$ | $1.03 \pm 0.08$ |
| | 50 | $1.05 \pm 0.05$ | $1.01 \pm 0.03$ | $1.02 \pm 0.06$ | $0.98 \pm 0.03$ | $1.05 \pm 0.03$ | $\mathbf{0.93} \pm 0.05$ | $0.94 \pm 0.07$ |
| | 100 | $1.05 \pm 0.1$ | $0.94 \pm 0.03$ | $0.91 \pm 0.04$ | $0.94 \pm 0.04$ | $1.01 \pm 0.05$ | $\mathbf{0.9} \pm 0.07$ | $0.93 \pm 0.08$ |
| *Humanoid* | 10 | $8.79 \pm 1.17$ | $8.26 \pm 1.91$ | $8.6 \pm 2.43$ | $10.97 \pm 0.15$ | $10.69 \pm 1.05$ | $\mathbf{6.82} \pm 1.34$ | $7.5 \pm 1.63$ |
| | 25 | $6.97 \pm 1.48$ | $6.59 \pm 1.81$ | $6.78 \pm 3.24$ | $11.01 \pm 0.17$ | $10.65 \pm 0.81$ | $\mathbf{5.11} \pm 1.34$ | $5.3 \pm 1.51$ |
| | 50 | $7.0 \pm 1.34$ | $6.42 \pm 1.55$ | $7.87 \pm 2.92$ | $11.07 \pm 0.25$ | $10.94 \pm 0.58$ | $4.27 \pm 0.94$ | $\mathbf{4.08} \pm 0.8$ |
| | 100 | $6.59 \pm 1.54$ | $5.33 \pm 1.2$ | $7.31 \pm 3.11$ | $11.01 \pm 0.23$ | $10.71 \pm 0.43$ | $3.4 \pm 0.48$ | $\mathbf{3.37} \pm 0.44$ |

◻ $p < 0.05$  ◻ $p < 0.01$  ◻ $p < 0.001$

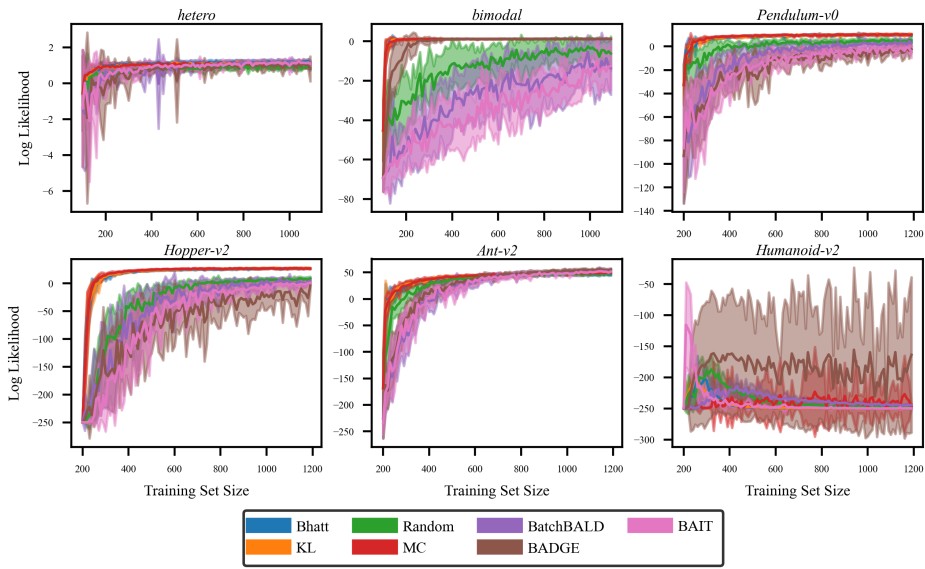

Figure 9: Mean Log Likelihood on test set as data was added to the training sets for Nflows Base.

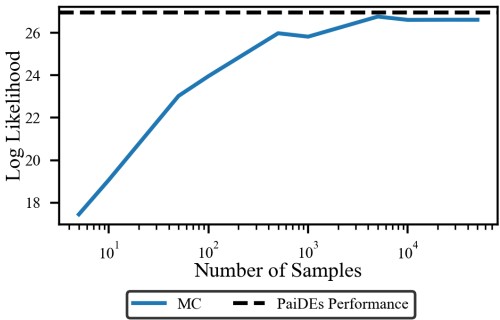

Figure 10: Log-Likelihood on the test set at the $100^{th}$ acquisition batch of the MC estimator on the *Hopper* environment as the number of samples increases, $K$, for Nflows Base. Experiment run across 10 seeds and the mean is being reported.

Table 8: Log Likelihood of a held out test set during training at different acquisition batches for Nflows Base. Experiments were across ten different seeds and the results are expressed as mean plus minus one standard deviation.

| Env | Acq. Batch | Random | BatchBALD | BADGE | BAIT | MC (BALD) | KL (ours) | Bhatt (ours) |
|---|---|---|---|---|---|---|---|---|
| | 10 | $0.54 \pm 0.27$ | $0.21 \pm 1.04$ | $-0.67 \pm 2.48$ | $0.27 \pm 0.7$ | $0.95 \pm 0.12$ | $0.93 \pm 0.14$ | $\mathbf{0.99} \pm 0.12$ |
| hetero | 25 | $0.79 \pm 0.18$ | $0.72 \pm 0.37$ | $0.77 \pm 0.26$ | $0.66 \pm 0.47$ | $1.06 \pm 0.06$ | $1.06 \pm 0.12$ | $\mathbf{1.11} \pm 0.09$ |
| | 50 | $0.8 \pm 0.18$ | $0.99 \pm 0.17$ | $0.95 \pm 0.18$ | $0.92 \pm 0.25$ | $1.08 \pm 0.04$ | $1.07 \pm 0.16$ | $\mathbf{1.15} \pm 0.14$ |
| | 100 | $0.86 \pm 0.14$ | $1.11 \pm 0.15$ | $0.8 \pm 0.7$ | $1.1 \pm 0.14$ | $1.09 \pm 0.1$ | $1.12 \pm 0.1$ | $\mathbf{1.14} \pm 0.13$ |
| | 10 | $-30.57 \pm 12.61$ | $-53.07 \pm 7.95$ | $-5.87 \pm 8.38$ | $-57.7 \pm 11.26$ | $1.08 \pm 0.09$ | $\mathbf{1.11} \pm 0.1$ | $\mathbf{1.11} \pm 0.1$ |
| bimodal | 25 | $-16.55 \pm 9.04$ | $-40.63 \pm 18.86$ | $1.0 \pm 0.33$ | $-41.4 \pm 14.8$ | $1.21 \pm 0.09$ | $1.17 \pm 0.1$ | $\mathbf{1.27} \pm 0.09$ |
| | 50 | $-10.85 \pm 8.5$ | $-22.17 \pm 13.04$ | $1.21 \pm 0.15$ | $-32.98 \pm 13.61$ | $\mathbf{1.22} \pm 0.11$ | $1.2 \pm 0.1$ | $1.21 \pm 0.11$ |
| | 100 | $-6.19 \pm 8.66$ | $-13.54 \pm 13.5$ | $\mathbf{1.31} \pm 0.16$ | $-12.65 \pm 11.91$ | $1.26 \pm 0.13$ | $1.26 \pm 0.1$ | $1.26 \pm 0.14$ |
| | 10 | $-5.55 \pm 6.8$ | $-47.16 \pm 31.67$ | $-53.04 \pm 26.71$ | $-61.82 \pm 38.52$ | $5.81 \pm 3.29$ | $\mathbf{6.17} \pm 1.86$ | $5.8 \pm 1.41$ |
| Pendulum | 25 | $-0.78 \pm 5.92$ | $-12.34 \pm 16.57$ | $-28.0 \pm 16.65$ | $-22.15 \pm 8.47$ | $\mathbf{8.72} \pm 0.45$ | $8.45 \pm 0.55$ | $8.3 \pm 0.7$ |
| | 50 | $3.77 \pm 1.8$ | $-2.51 \pm 5.06$ | $-7.43 \pm 6.72$ | $-8.46 \pm 4.94$ | $9.77 \pm 0.36$ | $\mathbf{9.95} \pm 0.5$ | $9.59 \pm 0.27$ |
| | 100 | $5.19 \pm 2.45$ | $3.04 \pm 2.38$ | $-2.29 \pm 2.9$ | $-4.41 \pm 5.31$ | $10.16 \pm 0.7$ | $\mathbf{10.49} \pm 0.23$ | $9.58 \pm 0.8$ |
| | 10 | $-136.85 \pm 53.59$ | $-147.16 \pm 77.99$ | $-120.43 \pm 77.99$ | $-201.21 \pm 50.44$ | $\mathbf{13.94} \pm 5.54$ | $11.59 \pm 4.32$ | $11.44 \pm 2.31$ |
| Hopper | 25 | $-28.0 \pm 28.16$ | $-94.48 \pm 71.55$ | $-118.09 \pm 65.33$ | $-88.87 \pm 71.52$ | $\mathbf{23.18} \pm 1.41$ | $22.71 \pm 0.86$ | $22.29 \pm 0.95$ |
| | 50 | $-0.55 \pm 6.82$ | $-19.33 \pm 25.02$ | $-38.16 \pm 25.17$ | $-17.2 \pm 9.63$ | $\mathbf{26.43} \pm 1.37$ | $25.72 \pm 1.04$ | $25.0 \pm 1.6$ |
| | 100 | $8.67 \pm 3.39$ | $4.53 \pm 5.31$ | $-4.16 \pm 6.1$ | $1.06 \pm 3.61$ | $26.62 \pm 1.14$ | $\mathbf{26.96} \pm 1.11$ | $26.88 \pm 0.91$ |
| | 10 | $-0.5 \pm 16.68$ | $-80.2 \pm 46.22$ | $-44.85 \pm 20.71$ | $-72.13 \pm 37.99$ | $24.48 \pm 5.88$ | $\mathbf{25.41} \pm 5.08$ | $21.59 \pm 9.0$ |
| Ant | 25 | $31.08 \pm 5.4$ | $17.06 \pm 13.16$ | $21.22 \pm 10.61$ | $23.12 \pm 11.37$ | $39.77 \pm 3.34$ | $\mathbf{40.62} \pm 2.85$ | $37.58 \pm 2.37$ |
| | 50 | $43.37 \pm 1.85$ | $43.43 \pm 4.81$ | $40.05 \pm 6.86$ | $40.89 \pm 7.56$ | $\mathbf{45.67} \pm 1.76$ | $45.07 \pm 1.62$ | $44.77 \pm 1.39$ |
| | 100 | $47.52 \pm 1.36$ | $54.38 \pm 3.23$ | $\mathbf{55.48} \pm 1.72$ | $51.56 \pm 3.68$ | $49.29 \pm 0.78$ | $48.12 \pm 2.05$ | $46.52 \pm 2.42$ |
| | 10 | $-200.61 \pm 20.02$ | $-243.1 \pm 6.03$ | $\mathbf{-182.46} \pm 99.5$ | $-230.61 \pm 9.92$ | $-249.1 \pm 0.69$ | $-221.09 \pm 10.68$ | $-201.28 \pm 22.78$ |
| Humanoid | 25 | $-223.54 \pm 13.72$ | $-221.22 \pm 9.42$ | $\mathbf{-173.12} \pm 89.96$ | $-245.86 \pm 1.8$ | $-247.84 \pm 1.74$ | $-246.29 \pm 1.92$ | $-247.16 \pm 1.04$ |
| | 50 | $-244.15 \pm 1.52$ | $-236.24 \pm 5.87$ | $\mathbf{-192.07} \pm 77.55$ | $-248.41 \pm 0.93$ | $-247.76 \pm 2.11$ | $-249.39 \pm 0.43$ | $-249.57 \pm 0.62$ |
| | 100 | $-247.82 \pm 1.28$ | $-244.58 \pm 2.1$ | $\mathbf{-163.78} \pm 124.9$ | $-249.71 \pm 0.32$ | $-246.47 \pm 2.99$ | $-249.76 \pm 0.4$ | $-249.97 \pm 0.05$ |

🟦 $p < 0.05$    🟧 $p < 0.01$    🟩 $p < 0.001$

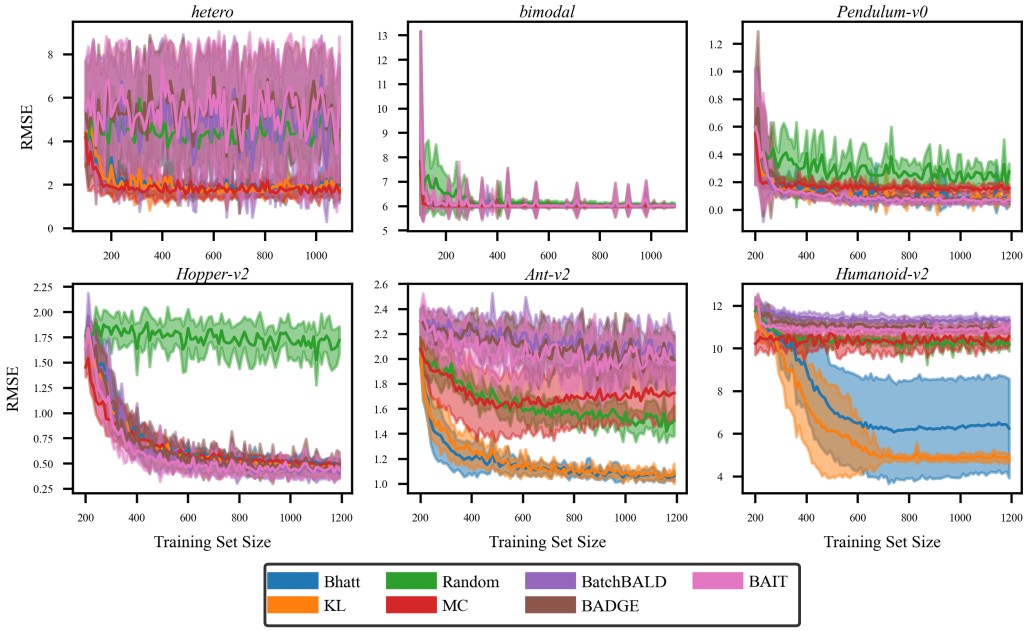

Figure 12: Mean RMSE on test set as data was added to the training sets for PNEs.

# E  Probabilistic Network Ensembles

In addition to ensembles of normalizing flows, we also explored the use of probabilistic network ensembles (PNEs) as a common approach for capturing uncertainty [12, 38, 39]. The PNEs were constructed by employing fixed dropout masks, where each ensemble component modeled a Gaussian distribution. The models were trained using negative log likelihood, with weights randomly initialized and bootstrapped samples from the training set to create diversity amongst the components. Our findings paralleled those of Nflows Base, in that PairEpEsts performed similarly or better than baselines and statistically significantly in higher dimensions. These results are presented in Figure 12 and Figure 13, as well as Table 9 and Table 10. Note that PNEs did not perform as well as Nflows Base as they are not as expressive this can be seen for the *bimodal* setting in Figure 11. Furthermore, we have included the 1D graphs illustrating the performance of PNEs in *hetero* and *bimodal* in Figure 11. Similarly to before, we also provide a comparison of the MC estimator as the number of samples increased in Figure 14. While we could have implemented ensembles with different output distributions to better fit the data, we decided to stick with Gaussians as they seem to be the most frequent choice [12, 38, 39]. Moreover, in order to pick the best

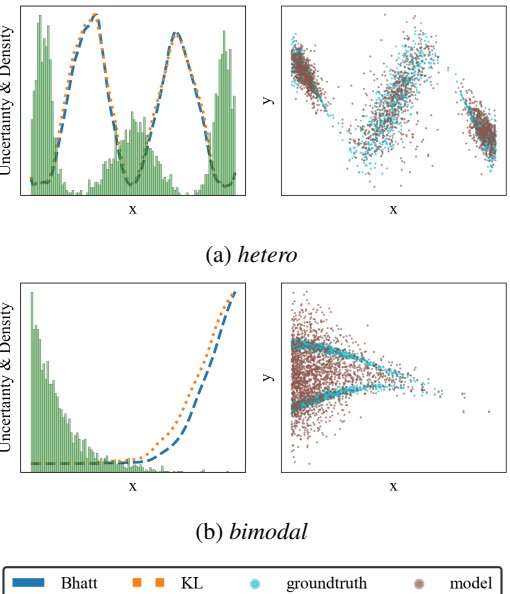

(a) *hetero*

(b) *bimodal*

Figure 11: In the right graphs, the cyan dots are the ground-truth data and the brown dots are sampled from PNEs. The 2 lines depict epistemic uncertainty corresponding to different estimators. The left graphs depicts the ground-truth data as the blue dots and its corresponding density as the orange histogram. Note the legend refers to the lines in the right graphs.

distributional fit would require a hyper-parameter search or apriori knowledge whereas NFs will learn the best distributional fit.

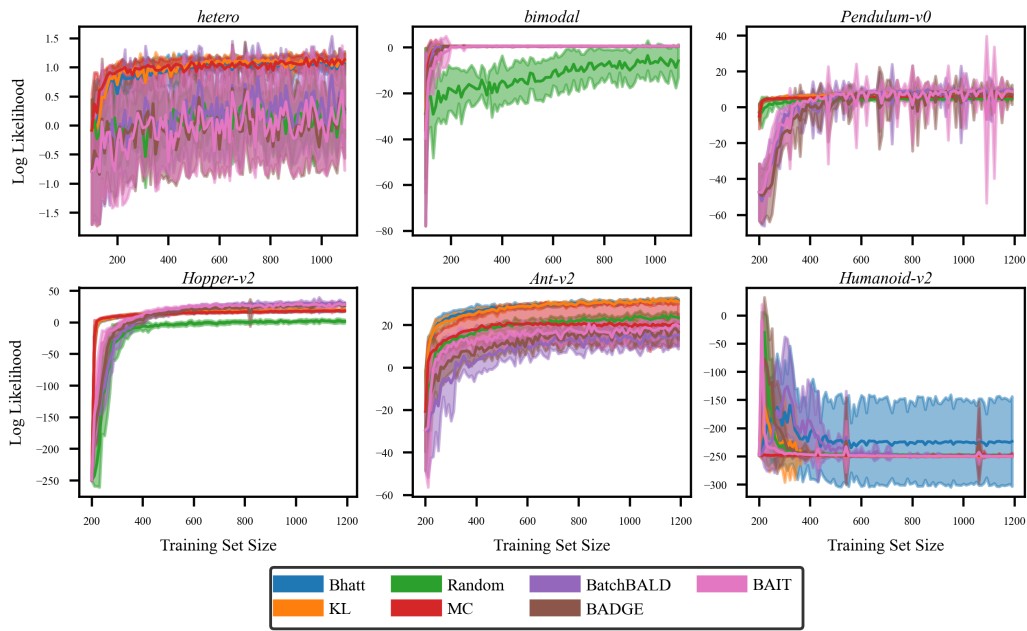

Figure 13: Mean Log Likelihood on the test set as data was added to the training sets for PNEs.

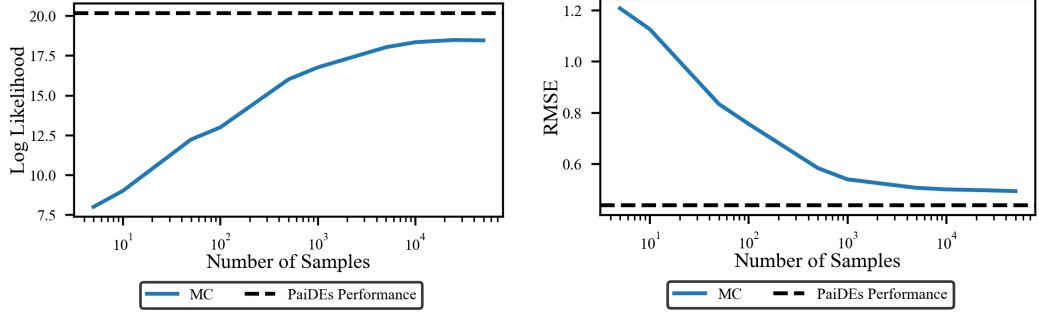

Figure 14: RMSE and Log-Likelihood on the test set at the $100^{th}$ acquisition batch of the MC estimator on the *Hopper* environment as the number of samples increases, $K$, for PNEs. Experiment run across 10 seeds and the mean is being reported.

Table 9: Mean RMSE on the test set for certain Acquisition Batches for PNEs. Experiments were across ten different seeds and the results are expressed as mean plus minus one standard deviation. Note that nothing was bolded for *bimodal* as many methods performed similarly.

| Env | Acq. Batch | Random | BatchBALD | BADGE | BAIT | MC (BALD) | KL (ours) | Bhatt (ours) |
|---|---|---|---|---|---|---|---|---|
| *hetero* | 10 | $3.9 \pm 1.47$ | $6.9 \pm 1.15$ | $6.91 \pm 1.77$ | $6.9 \pm 1.58$ | $\mathbf{1.94 \pm 0.34}$ | $2.11 \pm 0.34$ | $2.36 \pm 0.75$ |
| | 25 | $4.42 \pm 1.43$ | $4.43 \pm 2.81$ | $4.06 \pm 2.58$ | $5.02 \pm 2.91$ | $\mathbf{1.75 \pm 0.27}$ | $1.86 \pm 0.39$ | $1.96 \pm 0.44$ |
| | 50 | $3.96 \pm 1.71$ | $4.06 \pm 2.64$ | $5.12 \pm 2.78$ | $5.22 \pm 2.93$ | $1.78 \pm 0.51$ | $1.68 \pm 0.34$ | $\mathbf{1.67 \pm 0.27}$ |
| | 100 | $4.28 \pm 1.45$ | $5.1 \pm 2.92$ | $4.54 \pm 2.89$ | $6.01 \pm 2.82$ | $\mathbf{1.66 \pm 0.31}$ | $1.83 \pm 0.51$ | $1.98 \pm 0.81$ |
| *bimodal* | 10 | $6.52 \pm 0.88$ | $6.05 \pm 0.07$ | $6.19 \pm 0.27$ | $6.11 \pm 0.15$ | $6.01 \pm 0.04$ | $6.02 \pm 0.04$ | $6.01 \pm 0.04$ |
| | 25 | $6.1 \pm 0.08$ | $6.26 \pm 0.75$ | $6.31 \pm 0.83$ | $6.3 \pm 0.88$ | $6.01 \pm 0.04$ | $6.01 \pm 0.04$ | $6.01 \pm 0.04$ |
| | 50 | $6.13 \pm 0.13$ | $6.04 \pm 0.07$ | $6.02 \pm 0.03$ | $6.01 \pm 0.04$ | $6.01 \pm 0.04$ | $6.01 \pm 0.04$ | $6.01 \pm 0.04$ |
| | 100 | $6.06 \pm 0.08$ | $6.01 \pm 0.03$ | $6.03 \pm 0.03$ | $6.01 \pm 0.04$ | $6.01 \pm 0.04$ | $6.01 \pm 0.04$ | $6.01 \pm 0.04$ |
| *Pendulum* | 10 | $0.31 \pm 0.11$ | $0.15 \pm 0.02$ | $0.16 \pm 0.05$ | $\mathbf{0.15 \pm 0.03}$ | $0.19 \pm 0.04$ | $0.18 \pm 0.06$ | $0.2 \pm 0.07$ |
| | 25 | $0.27 \pm 0.09$ | $\mathbf{0.11 \pm 0.03}$ | $0.12 \pm 0.03$ | $\mathbf{0.11 \pm 0.03}$ | $0.16 \pm 0.02$ | $0.12 \pm 0.04$ | $0.13 \pm 0.04$ |
| | 50 | $0.25 \pm 0.07$ | $\mathbf{0.08 \pm 0.02}$ | $0.09 \pm 0.02$ | $\mathbf{0.08 \pm 0.02}$ | $0.15 \pm 0.03$ | $\mathbf{0.08 \pm 0.02}$ | $0.09 \pm 0.03$ |
| | 100 | $0.28 \pm 0.06$ | $\mathbf{0.07 \pm 0.03}$ | $0.07 \pm 0.02$ | $\mathbf{0.07 \pm 0.02}$ | $0.16 \pm 0.03$ | $0.09 \pm 0.06$ | $0.08 \pm 0.05$ |
| *Hopper* | 10 | $1.85 \pm 0.14$ | $1.44 \pm 0.31$ | $1.12 \pm 0.29$ | $1.16 \pm 0.37$ | $\mathbf{0.94 \pm 0.16}$ | $1.19 \pm 0.21$ | $1.14 \pm 0.18$ |
| | 25 | $1.9 \pm 0.14$ | $0.6 \pm 0.13$ | $0.66 \pm 0.19$ | $\mathbf{0.55 \pm 0.12}$ | $0.73 \pm 0.13$ | $0.72 \pm 0.08$ | $0.71 \pm 0.1$ |
| | 50 | $1.83 \pm 0.13$ | $0.54 \pm 0.15$ | $0.55 \pm 0.15$ | $\mathbf{0.51 \pm 0.12}$ | $0.57 \pm 0.06$ | $0.57 \pm 0.06$ | $0.57 \pm 0.05$ |
| | 100 | $1.72 \pm 0.14$ | $\mathbf{0.4 \pm 0.05}$ | $0.49 \pm 0.13$ | $0.43 \pm 0.05$ | $0.5 \pm 0.04$ | $0.44 \pm 0.04$ | $0.49 \pm 0.05$ |
| *Ant* | 10 | $1.89 \pm 0.12$ | $2.27 \pm 0.13$ | $2.24 \pm 0.12$ | $2.15 \pm 0.18$ | $1.87 \pm 0.25$ | $1.62 \pm 0.16$ | $\mathbf{1.52 \pm 0.18}$ |
| | 25 | $1.74 \pm 0.13$ | $2.15 \pm 0.21$ | $2.14 \pm 0.18$ | $2.12 \pm 0.25$ | $1.67 \pm 0.25$ | $1.43 \pm 0.12$ | $\mathbf{1.39 \pm 0.12}$ |
| | 50 | $1.55 \pm 0.1$ | $2.2 \pm 0.13$ | $2.16 \pm 0.14$ | $2.11 \pm 0.18$ | $1.61 \pm 0.22$ | $\mathbf{1.29 \pm 0.06}$ | $1.3 \pm 0.04$ |
| | 100 | $1.5 \pm 0.12$ | $2.01 \pm 0.2$ | $1.99 \pm 0.2$ | $1.97 \pm 0.15$ | $1.73 \pm 0.22$ | $1.26 \pm 0.08$ | $\mathbf{1.24 \pm 0.04}$ |
| *Humanoid* | 10 | $10.73 \pm 0.32$ | $11.63 \pm 0.19$ | $11.22 \pm 0.29$ | $10.95 \pm 0.17$ | $10.41 \pm 0.53$ | $\mathbf{9.95 \pm 1.1}$ | $10.65 \pm 0.84$ |
| | 25 | $10.38 \pm 0.22$ | $11.36 \pm 0.26$ | $11.07 \pm 0.26$ | $10.91 \pm 0.25$ | $10.44 \pm 0.55$ | $\mathbf{6.54 \pm 2.13}$ | $8.22 \pm 2.47$ |
| | 50 | $10.29 \pm 0.19$ | $11.37 \pm 0.3$ | $10.98 \pm 0.22$ | $10.9 \pm 0.25$ | $10.34 \pm 0.81$ | $\mathbf{4.91 \pm 0.24}$ | $6.33 \pm 2.24$ |
| | 100 | $10.17 \pm 0.3$ | $11.32 \pm 0.16$ | $10.99 \pm 0.26$ | $11.05 \pm 0.21$ | $10.57 \pm 0.41$ | $\mathbf{4.84 \pm 0.24}$ | $6.25 \pm 2.32$ |

$p < 0.05$    $p < 0.01$    $p < 0.001$

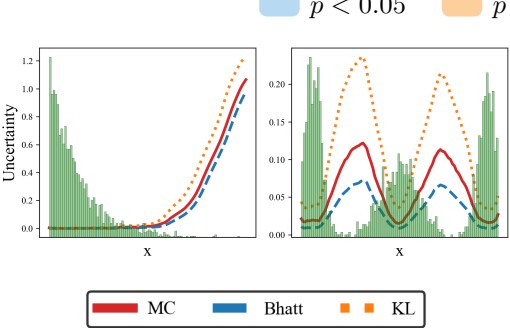

MC —— Bhatt —— KL ▪ ▪ ▪

Figure 15: The bias introduced by PairEpEsts for PNEs compared to the MC method on the 1D environments.

Table 10: Log Likelihood on the test set during training at different acquisition batches for PNEs. Experiments were across ten different seeds and the results are expressed as mean plus minus one standard deviation. Note that no values were highlighted for Hopper despite the results being statistically significant. In this case, the results are statistically significantly worse and we found it misleading to highlight.

| Env | Acq. Batch | Random | BatchBALD | BADGE | BAIT | MC (BALD) | KL (ours) | Bhatt (ours) |
|---|---|---|---|---|---|---|---|---|
| *hetero* | 10 | $0.13 \pm 0.57$ | $-0.67 \pm 0.55$ | $-0.72 \pm 0.61$ | $-0.78 \pm 0.59$ | $\mathbf{0.89 \pm 0.1}$ | $0.87 \pm 0.15$ | $0.74 \pm 0.3$ |
| | 25 | $0.03 \pm 0.47$ | $0.33 \pm 0.76$ | $0.38 \pm 0.76$ | $0.13 \pm 0.82$ | $\mathbf{1.02 \pm 0.11}$ | $0.95 \pm 0.14$ | $0.92 \pm 0.24$ |
| | 50 | $0.23 \pm 0.57$ | $0.48 \pm 0.71$ | $0.14 \pm 0.77$ | $0.11 \pm 0.85$ | $1.01 \pm 0.15$ | $\mathbf{1.1 \pm 0.11}$ | $1.09 \pm 0.12$ |
| | 100 | $0.23 \pm 0.42$ | $0.23 \pm 0.78$ | $0.28 \pm 0.85$ | $-0.03 \pm 0.74$ | $\mathbf{1.1 \pm 0.11}$ | $1.11 \pm 0.17$ | $1.08 \pm 0.2$ |
| *bimodal* | 10 | $-17.71 \pm 10.01$ | $0.32 \pm 0.33$ | $0.41 \pm 0.21$ | $-1.36 \pm 4.14$ | $0.6 \pm 0.1$ | $0.59 \pm 0.14$ | $\mathbf{0.61 \pm 0.07}$ |
| | 25 | $-20.78 \pm 5.64$ | $0.61 \pm 0.19$ | $0.48 \pm 0.23$ | $0.57 \pm 0.21$ | $0.66 \pm 0.03$ | $0.66 \pm 0.03$ | $\mathbf{0.67 \pm 0.04}$ |
| | 50 | $-12.93 \pm 8.55$ | $0.66 \pm 0.09$ | $0.59 \pm 0.16$ | $0.61 \pm 0.18$ | $\mathbf{0.69 \pm 0.02}$ | $\mathbf{0.69 \pm 0.03}$ | $\mathbf{0.69 \pm 0.03}$ |
| | 100 | $-5.79 \pm 7.19$ | $0.66 \pm 0.1$ | $0.64 \pm 0.11$ | $0.68 \pm 0.08$ | $\mathbf{0.7 \pm 0.02}$ | $\mathbf{0.7 \pm 0.02}$ | $\mathbf{0.7 \pm 0.02}$ |
| *Pendulum* | 10 | $2.35 \pm 2.94$ | $-10.76 \pm 7.28$ | $-15.25 \pm 8.86$ | $-9.76 \pm 9.09$ | $4.94 \pm 0.48$ | $\mathbf{5.51 \pm 0.72}$ | $5.21 \pm 0.61$ |
| | 25 | $4.19 \pm 0.89$ | $-0.4 \pm 14.08$ | $0.2 \pm 13.03$ | $4.68 \pm 2.98$ | $5.45 \pm 0.43$ | $6.21 \pm 0.98$ | $\mathbf{6.23 \pm 0.84}$ |
| | 50 | $4.34 \pm 0.75$ | $8.34 \pm 0.83$ | $7.97 \pm 0.74$ | $\mathbf{8.68 \pm 0.72}$ | $5.92 \pm 0.32$ | $7.65 \pm 0.5$ | $7.15 \pm 0.76$ |
| | 100 | $4.69 \pm 0.56$ | $\mathbf{9.53 \pm 0.96}$ | $7.07 \pm 5.49$ | $8.8 \pm 1.23$ | $6.09 \pm 0.42$ | $7.66 \pm 0.64$ | $7.77 \pm 0.68$ |
| *Hopper* | 10 | $-31.94 \pm 28.24$ | $-26.65 \pm 15.57$ | $-12.96 \pm 9.55$ | $-10.81 \pm 4.25$ | $\mathbf{9.36 \pm 1.79}$ | $8.55 \pm 1.34$ | $8.55 \pm 1.88$ |
| | 25 | $-5.74 \pm 2.66$ | $12.62 \pm 4.32$ | $13.88 \pm 2.27$ | $\mathbf{17.29 \pm 3.45}$ | $13.31 \pm 2.3$ | $13.9 \pm 1.3$ | $13.93 \pm 1.6$ |
| | 50 | $-2.31 \pm 3.08$ | $27.28 \pm 2.42$ | $23.49 \pm 2.97$ | $\mathbf{28.08 \pm 2.76}$ | $16.43 \pm 1.27$ | $17.6 \pm 1.89$ | $16.93 \pm 1.43$ |
| | 100 | $1.75 \pm 2.83$ | $30.35 \pm 1.44$ | $26.54 \pm 2.19$ | $\mathbf{28.44 \pm 1.52}$ | $18.5 \pm 0.65$ | $20.19 \pm 1.37$ | $18.54 \pm 1.23$ |
| *Ant* | 10 | $11.83 \pm 2.29$ | $-4.91 \pm 12.37$ | $3.29 \pm 6.43$ | $10.2 \pm 5.57$ | $13.8 \pm 5.87$ | $22.13 \pm 3.11$ | $\mathbf{23.32 \pm 2.73}$ |
| | 25 | $17.78 \pm 2.28$ | $5.79 \pm 7.68$ | $8.44 \pm 3.68$ | $11.97 \pm 5.41$ | $18.69 \pm 9.06$ | $26.98 \pm 1.64$ | $\mathbf{27.66 \pm 1.78}$ |
| | 50 | $22.21 \pm 2.11$ | $12.24 \pm 3.25$ | $14.94 \pm 3.3$ | $16.16 \pm 3.2$ | $20.54 \pm 8.88$ | $\mathbf{29.84 \pm 1.09}$ | $29.62 \pm 0.7$ |
| | 100 | $23.4 \pm 1.85$ | $15.26 \pm 6.5$ | $16.87 \pm 3.75$ | $19.6 \pm 2.07$ | $20.03 \pm 9.87$ | $31.24 \pm 1.09$ | $\mathbf{31.63 \pm 0.9}$ |
| *Humanoid* | 10 | $-230.41 \pm 19.87$ | $\mathbf{-156.59 \pm 82.79}$ | $-210.52 \pm 53.23$ | $-242.95 \pm 2.96$ | $-247.83 \pm 1.71$ | $-226.42 \pm 46.74$ | $-194.78 \pm 73.74$ |
| | 25 | $-245.45 \pm 2.24$ | $\mathbf{-224.95 \pm 37.12}$ | $-245.82 \pm 5.03$ | $-247.3 \pm 1.26$ | $-247.83 \pm 1.19$ | $-247.39 \pm 3.18$ | $-232.34 \pm 42.02$ |
| | 50 | $-246.91 \pm 1.28$ | $-247.38 \pm 3.28$ | $-249.28 \pm 0.52$ | $-249.2 \pm 0.52$ | $-248.06 \pm 1.82$ | $-249.41 \pm 0.73$ | $\mathbf{-223.49 \pm 77.83}$ |
| | 100 | $-247.74 \pm 0.6$ | $-248.97 \pm 1.1$ | $-249.4 \pm 0.6$ | $-249.66 \pm 0.53$ | $-247.25 \pm 3.36$ | $-249.91 \pm 0.12$ | $\mathbf{-223.29 \pm 79.8}$ |

■ $p < 0.05$   ■ $p < 0.01$   ■ $p < 0.001$

# F   Introduction to Normalizing Flows

NFs are powerful non-parametric models that have demonstrated the ability to fit flexible multi-modal distributions [54, 53]. These models achieve this by transforming a simple base continuous distribution, such as Gaussian or Beta, into a more complex one using the change of variable formula. By enabling scoring and sampling from the fitted distribution, NFs find application across various problem domains. Let $B$ represent the base distribution, a D-dimensional continuous random vector with $p_B(b)$ as its density function, and let $Y = g(B)$, where $g$ is an invertible function with an existing inverse $g^{-1}$, and both $g$ and $g^{-1}$ are differentiable. Leveraging the change of variable formula, we can express the distribution of $Y$ as follows:

$$p_Y(y) = p_B(g^{-1}(y))|\det(J(g^{-1}(y)))|, \tag{11}$$

where $J(\cdot)$ denotes the Jacobian, and $\det$ signifies the determinant. The first term on the right-hand side of Equation 11 governs the shape of the distribution, while $|\det(J(g^{-1}(y)))|$ normalizes it, ensuring the distribution integrates to one. Complex distributions can be effectively modeled by making $g(b)$ a learnable function with tunable parameters $\theta$, denoted as $g_\theta(b)$. However, it is essential to select $g$ carefully to guarantee its invertibility and differentiability. For examples of suitable choices, please refer to [47].

# G   Hypothesis Testing Details

We conducted Welch's t-tests to compare means $(\mu_i, \mu_j)$ between different estimators, as this test relaxes the assumption of equal variances compared to other hypothesis tests [13]. The means for both KL and Bhatt were compared to each of the baseline methods: BatchBALD, BADGE, BAIT, MC and random. To control the family-wise error rate (FWER), we performed a Holm-Bonferroni correction across each setting, environment, and acquisition batch. This follows best practices to ensure our results are statistically significant and do not occur just by random chance.

