# OpenReview forum: "Epistemic Uncertainty Estimation in Regression Ensemble Models with Pairwise Epistemic Estimators"
_NeurIPS.cc/2025/Conference — NeurIPS 2025 poster_

### Official Review · Reviewer_yg5h · 2025-06-23

**Clarity:** 3
**Significance:** 2
**Originality:** 3
**Rating:** 4
**Confidence:** 4

**Summary:**

The paper utilizes Paired Distance Estimators to efficiently measure epistemic uncertainty in regression ensemble models, i.e. estimating the differential entropy of the mixture distribution.
Specifically, two practical implementations based on the Bhattacharyya distance and KL divergence are introduced.
They demonstrate the performance of this approach on two synthetic tasks and for active learning on real-world datasets for an ensemble of NFlow base models.

**Questions:**

- How does PairEpEst-KL relate to the Expected Pairwise KL-divergence discussed in Malinin et al. 2021 and Schweighofer et al. 2023 as measure of epistemic uncertainty in the classification setting? Is it an extention to the regression setting or is there a fundamental difference between those quantities?

- Given the multimodel output distribution of an ensemble of NFlows base, how was the point prediction for calculating the RMSE obtained?

- Why not reporting the MC estimate in Figure 2, when this is done in Figure 8 in the appendix?
Also, why does the estimated uncertainty look different in the two figures?

- Why are many different colors introduced for different significant levels in Table 1, when there are only two entries with p < 0.01 and no entries that are significant for p < 0.05 and p < 0.001?

- Why was a uniform weighting of components, i.e. $\pi_j = 1/M$ (line 198) chosen? Are there alternative ways to weight ensemble members that could be interesting?


Malinin, A., & Gales, M. (2021). Uncertainty estimation in autoregressive structured prediction.

Schweighofer, K., Aichberger, L., Ielanskyi, M., & Hochreiter, S. (2023). Introducing an improved information-theoretic measure of predictive uncertainty.

**Ethical Concerns:**

["NO or VERY MINOR ethics concerns only"]

**Final Justification:**

As detailed in my response to the rebuttal, many of my points of critique have been adequately addressed.
However, the presentation in Figure 2 is still not comprehendable to me, I would go with the presentation as in Figure 8.
Furthermore, my doubts regarding the effectiveness of the method beyond the humanoid task have not been resolved.
In principle, also efficiency (i.e. runtime) could have been the major metric, controlling only that performance is comparable.
However, the current presentation in Table 2 focus entirely on performance, where PaiDEs perform significantly better on humanoid, but not on other tasks.

Overall, I found the paper very interesting and think its merits would warrant acceptance, yet the experimental evalution or at least its presentation is on the weaker side and could be improved further.

**Limitations:**

I don't agree that the stated limitation of introducing bias in the estimation of epistemic uncertainty (line 287) does not affect the relative ordering of points.
The bias is input dependent as can also be seen in Figure 8, thus might change the relative ordering of individual inputs for active learning, unless I missed some technical result on the monotonicity of the bias depending on $H(y|x)$?
Nevertheless, it could be that the statement is true in practice, but I would like to see an experiment looking at ranking correlations between PairEpEsts and the MC estimate with a very high number of samples (to rule out MC variance) on the ordering of inputs on one of the tasks to confirm this.

**Paper Formatting Concerns:**

Excessive use of negative \vspace under Figure 4. The figure description and main text nearly overlap each other.

**Quality:**

3

**Strengths And Weaknesses:**

## Strengths

- Good motivation for why using PaiDEs is necessary for estimating epistemic uncertainty in regression ensemble models. While this is especially true for their utilized NF predictive model, this may also be a concern for usual PNEs.
- While the theoretical results are following the steps of Kolchinsky and Tracey and are not novel per se, they are relevant and interesting for the uncertainty estimation community in machine learning.

## Weaknesses

- The experimental results are unfortunately not very encouraging. The synthetic experiments look reasonable, but the active learning results only look convincing on the Humanoid task. Also adding results for the full dataset would put the active learning results better into perspective.
- Figure 3 only reports the difference between means. Having variances or CI would put the differences between PaiDEs and MC estimation better into perspective.
- The runtime issue of MC estimation is a bit unclear to me. Is this a general issue or intrinsic to an ensemble of NFlows base models? Because I don't see how sampling would be so much slower when given the output distributions of members of the PNE. Also, I did not find a time comparison for PNEs.

---

> ### Author Rebuttal · Authors · 2025-07-30
>
> Dear Reviewer,
>
> Thank you for your thoughtful review. We're glad you found the motivation for using PaiDEs to estimate epistemic uncertainty in regression ensembles compelling—particularly with Normalizing Flows, but also potentially for standard PNEs. We also appreciate your recognition that, while our theoretical results build on prior work by Kolchinsky and Tracey, they remain relevant and valuable to the uncertainty estimation community.
>
> **Legend:**
> **W** – Reviewer *Weakness*
> **R** – Author *Response*
> **Q** – Reviewer *Question*
> **A** – Author *Answer*
>
> **W1:** Limited effectiveness beyond Humanoid; need full-dataset performance
>
> **R1:** While the gains from PairEpEsts are most pronounced in the highest-dimensional environment, they consistently match or outperform baselines across all environments, model types, and acquisition batches (see Tables 2 and 4). This includes low-dimensional synthetic tasks and mid-dimensional control tasks such as Hopper and Ant.
>
> Crucially,  PairEpEsts deliver these improvements while being more computationally efficient and robust. We also include statistical significance testing (Table 1), which is often omitted in related work but is key to verifying that improvements are meaningful rather than due to noise.
>
> We believe the breadth, consistency, and statistical rigor of our evaluation make the results compelling.
>
> Regarding full-dataset performance: we agree this adds valuable perspective. Figure 4 and Figure 12 now include RMSEs using the full dataset. We reproduce the values below:
>
> | Model        | Hetero | Bimodal | Pendulum | Hopper | Ant  | Humanoid |
> |--------------|--------|---------|----------|--------|------|----------|
> | NFlows Base  | 1.4    | 6       | 0.02     | 0.21   | 0.85 | 2.97     |
> | PNEs         | 1.6    | 6       | 0.04     | 0.31   | 1.17 | 4.57     |
>
> **W2:** Figure 3 lacks variances or confidence intervals
>
> **R2:** We have now added mean ± standard deviation values in the table below and the paper to better reflect the variance in performance. These were initially omitted as the standard deviations were consistently small:
>
> | Samples | RMSE ± Std |
> |---------|------------|
> | 5       | 0.60 ± 0.09 |
> | 10      | 0.54 ± 0.05 |
> | 50      | 0.39 ± 0.05 |
> | 100     | 0.36 ± 0.03 |
> | 500     | 0.32 ± 0.03 |
> | 1000    | 0.31 ± 0.02 |
> | 5000    | 0.30 ± 0.02 |
> | 10000   | 0.29 ± 0.03 |
> | 20000   | 0.29 ± 0.02 |
> | 50000   | 0.30 ± 0.02 |
>
> **W3:** Clarify runtime cost of MC estimation and lack of PNE runtime
>
> **R3:** The runtime bottleneck arises primarily in active learning, where epistemic uncertainty must be estimated for every input in a large candidate pool (e.g., 1000 inputs per acquisition step). MC-based methods require drawing K = 5000 per candidate then estimating mutual information, which becomes expensive in high-dimensional output spaces like Humanoid.
>
> This cost scales with both the number of samples and the output dimensionality. As a result, MC estimation is especially burdensome under these settings.
>
> We do not include a separate runtime plot for PNEs because they share a similar architecture to the base distribution used in NFlows Base. Both models sample from Gaussians whose parameters are predicted by neural networks. As shown in Appendix A.10 of [1], computing epistemic uncertainty in the base distribution space is equivalent to the output space. Therefore, the runtime of PNEs is effectively reflected in Figures 5 and 6. We’ve clarified this in the manuscript.
>
> **Q1:** Relationship between PairEpEst-KL and EPKL.
>
> **A1:** PairEpEst is a general framework that extends beyond KL-divergence, allowing practitioners to apply a range of premetrics, such as KL divergence, Bhattacharyya distance, Hellinger distance, total variation distance, and Wasserstein distance. This generality is important when KL is undefined between ensemble members.
>
> In the specific case of KL divergence, we show that PairEpEst-KL provides a strictly tighter upper bound on mutual information than the EPKL estimator used in prior work [2,3]. This theoretical analysis has been added to the paper.
>
> We formally prove that:
>
> $\text{PairEpEst-KL} < \text{EPKL}$
>
> **Theorem:** Let $\( p_1(y), \ldots, p_M(y) \)$ be probability distributions. Define
>
> $A := \frac{1}{M(M-1)} \sum_{i \neq j} D_{\mathrm{KL}}(p_i(y) \|\| p_j(y)),$
>
> and
>
> $B := - \frac{1}{M} \sum_{i=1}^M \ln \left( \frac{1}{M} \sum_{j=1}^M \exp \big(- D_{\mathrm{KL}}(p_i(y) \|\| p_j(y)) \big) \right).$
>
> Then,
>
> $B \leq \frac{1}{M^2} \sum_{i,j} D_{\mathrm{KL}}(p_i(y) \|\| p_j(y)) < A,$
>
> and in particular,
>
> $B < A.$
>
> **Proof:** Set
>
> $s_{ij} := - D_{\mathrm{KL}}(p_i(y) \|\| p_j(y)) \leq 0.$
>
> By Jensen’s inequality applied to the concave function $\ln(\cdot)$, for each fixed $i$,
>
> $\ln \left( \frac{1}{M} \sum_{j=1}^M e^{s_{ij}} \right) \geq \frac{1}{M} \sum_{j=1}^M s_{ij}.$
>
>
> Multiplying both sides by -1 reverses the inequality:
>
> $- \ln \left( \frac{1}{M} \sum_{j=1}^M e^{s_{ij}} \right) \leq - \frac{1}{M} \sum_{j=1}^M s_{ij}.$
>
> Substituting back for $s_{ij}$,
>
> $- \ln \left( \frac{1}{M} \sum_{j=1}^M \exp\big(- D_{\mathrm{KL}}(p_i(y) \|\| p_j(y)) \big) \right) \leq \frac{1}{M} \sum_{j=1}^M D_{\mathrm{KL}}(p_i(y) \|\| p_j(y)).$
>
> Averaging over $i$, we get
>
> $B = \frac{1}{M} \sum_{i=1}^M \left[- \ln \left( \frac{1}{M} \sum_{j=1}^M e^{-D_{\mathrm{KL}}(p_i \|\| p_j)} \right) \right] \leq \frac{1}{M^2} \sum_{i=1}^M \sum_{j=1}^M D_{\mathrm{KL}}(p_i \|\| p_j).$
>
> Since $D_{\mathrm{KL}}(p_i \|\| p_i) = 0$, the sum over all $\(i,j\)$ equals the sum over $i\neq j$:
>
> $\sum_{i=1}^M \sum_{j=1}^M D_{\mathrm{KL}}(p_i \|\| p_j) = \sum_{i \neq j} D_{\mathrm{KL}}(p_i \|\| p_j).$
>
> Note that
>
> $M^2 > M(M-1) \implies \frac{1}{M^2} < \frac{1}{M(M-1)},$
>
> so
>
> $B \leq \frac{1}{M^2} \sum_{i \neq j} D_{\mathrm{KL}}(p_i \|\| p_j) < \frac{1}{M(M-1)} \sum_{i \neq j} D_{\mathrm{KL}}(p_i \|\| p_j) = A.$
>
> Hence,
>
> $B < A,$
>
> completing the proof.
>
> **Q2:** How point prediction was computed for RMSE for Nflows Base?
>
> **A2:** For each test input, we draw samples from the ensemble’s output distribution and compute the mean prediction. This mean is then compared to the ground-truth value to calculate RMSE.
>
> **Q3:** Why MC estimate is missing in Figure 2 and uncertainty appears different?
>
> **R3:** Figure 2 applies a min-max normalization to each estimator, scaling the values to the [0, 1] range for improved visual clarity. This makes it easier to interpret relative uncertainty and observe trends, such as lower uncertainty for points with fewer observations. For more details, please see our response to Limitations below.
>
> **Q4:** Color coding for significance in Table 1.
>
> **R4:** We included the full range of commonly reported significance levels in statistics to emphasize that none of the results reached the p < 0.05 or p < 0.001 thresholds. This also ensures consistency across the paper, as all colors are used in later tables where more significance levels are present.
>
> **Q5:** Why use uniform weighting of ensemble components?
>
> **R5:** We use uniform weighting because it is standard in ensemble learning and uncertainty estimation. While alternative weightings may be interesting (e.g., to induce certain multimodal properties), we are not aware of prior work exploring this in our context.
>
> **Limitations:** Does estimator bias affect active learning rankings?
>
> **R:** Thank you for this thoughtful point. To directly test whether bias alters active learning rankings, we computed Spearman’s rank correlation between our estimators (KL and Bhatt) and high-sample MC estimates on the data from Figure 8.
>
> Spearman’s coefficient measures monotonic agreement in rankings (1 = perfect correlation). Our results show values close to 1 across all settings, indicating that relative ordering is preserved. All p-values were extremely small (max p = 1.11e-83), confirming statistical significance.
>
> We have updated the manuscript to reflect this and revised the original statement to clarify that while bias exists, its impact on active learning ranking is negligible in practice. Not that the order is preserved exactly.
>
> This analysis has been added to the paper.
>
> | Model        | Dataset  | KL Corr | Bhatt Corr |
> |--------------|----------|---------|------------|
> | NFlows Base  | Bimodal  | 0.9958  | 0.9958     |
> | NFlows Base  | Hetero   | 0.9976  | 0.9986     |
> | PNEs         | Bimodal  | 0.9893  | 0.9893     |
> | PNEs         | Hetero   | 0.9943  | 0.9972     |
>
> **Formatting:** Excessive use of negative \vspace under Figure 4. The figure description and main text nearly overlap each other.
>
> **R:** Thank you for noticing, we have corrected this in the manuscript.
>
> **References**
>
> [1] Berry, L., & Meger, D. (2023). *Normalizing flow ensembles for rich aleatoric and epistemic uncertainty modeling*. arXiv preprint arXiv:2302.01312
>
> [2] Malinin, A., & Gales, M. (2020). *Uncertainty estimation in autoregressive structured prediction*. arXiv preprint arXiv:2002.07650
>
> [3] Schweighofer, K., Aichberger, L., Ielanskyi, M., & Hochreiter, S. (2023). *Introducing an improved information-theoretic measure of predictive uncertainty*. arXiv preprint arXiv:2311.08309

---

> > ### Comment · Reviewer_yg5h · 2025-08-01
> >
> > Thank you very much for the thorough rebuttal.
> >
> > ---
> > On the positive side:
> >
> > The provided theoretical analysis that PairEpEst-KL is an upper bound on EPKL is very interesting, thank you for providing that!
> >
> > Many points of my critique have been addressed, such as the bias in active learning rankings and the additional details regarding the experimental results, e.g. full dataset performance or confidence intervals.
> >
> > ---
> > On the negative side:
> >
> > I must confess I don't understand the answer to "Why MC estimate is missing in Figure 2 and uncertainty appears different?" Does that mean both are independently scaled to [0,1]? Why is this more interesting than the presentation in Figure 8? Also, I couldn't find more details in the **Limitations** section of the rebuttal as referred to by the authors.
> >
> > Furthermore, while it is commendable that significance is tested, PaiDEs only significantly perform better on humanoid according to Table 1. I agree that not every method needs to outperform prior methods, especially if its aim is to improve efficiency, but the way the results are presented, it feels a bit underwhelming. I would add runtimes as primary metric to this table, also such that future readers can understand the merits of the proposed method.
> >
> > ---
> > However, I am fine with the overall improvements and will raise my score to 4, reflecting my final assessment of the current status of this paper: borderline accept where reasons to accept outweight reasons to reject.

---

> > > ### Author Response · Authors · 2025-08-03
> > >
> > > Dear Reviewer,
> > >
> > > Thank you for your thoughtful comments and for your engagement throughout the review process. We also sincerely appreciate your decision to raise your score. Please note, your reviewer score no longer appears in our author console, so we kindly ask you to double-check that your updated score was submitted correctly. It has a significant impact on the paper. Apologies if this has already been done.
> > >
> > > **Regarding Figure 2 and uncertainty scaling:**
> > >
> > > Yes, both curves in Figure 2 are independently scaled to $[0,1]$, purely for visualization purposes. This does not affect the active learning process, as each estimator is used independently to acquire points—only the relative ranking within each estimator matters.
> > >
> > > The goal of Figure 2 is to illustrate that our method effectively captures epistemic uncertainty in regions with varying observational density, using a simple 1D case where the behavior of uncertainty estimates is easy to interpret. We omitted MC Dropout from this figure to avoid visual clutter. When scaled to $[0,1]$, the many overlapping curves resulted in unreadable plots.
> > >
> > > Figure 8, which presents estimator bias and unscaled uncertainty values, appears later in the paper. Figure 2 is placed earlier, near the introduction of active learning. We believe this structure improves the overall flow of the manuscript while ensuring that all key information is clearly presented and appropriately contextualized. Due to page limitations, some results had to be placed in the appendix. While we understand and appreciate your preference for a different organization, we emphasize that all important information is conveyed in the text.
> > >
> > > **On significance and runtimes:**
> > >
> > > We already include per-step acquisition runtimes in Figure 5, which we believe are more informative and broadly useful than total active learning runtimes. These per-step times isolate the cost of the acquisition function itself, allowing practitioners to better assess how our method would scale when used with different model architectures or training procedures.
> > > That said, we appreciate the value of end-to-end comparisons. In response to your suggestion, we’ve added the following two tables to the appendix, reporting total active learning runtimes (in minutes) across all environments.
> > >
> > > **NFlows Base**
> > >
> > > |         | Hetero | Bimodal | Pendulum | Hopper | Ant   | Humanoid |
> > > |---------|--------|---------|----------|--------|-------|----------|
> > > | KL      | 58.14  | 58.22   | 76.02    | 77.06  | 80.21 | 96.28    |
> > > | Bhatt   | 57.72  | 58.04   | 76.86    | 76.94  | 78.54 | 95.58    |
> > > | MC      | 61.87  | 61.82   | 94.44    | 98.66  | 117.4 | 155.34   |
> > >
> > > **PNEs**
> > >
> > > |         | Hetero | Bimodal | Pendulum | Hopper | Ant   | Humanoid |
> > > |---------|--------|---------|----------|--------|-------|----------|
> > > | KL      | 22.68  | 22.50   | 28.44    | 28.60  | 30.58 | 35.24    |
> > > | Bhatt   | 22.50  | 22.36   | 28.24    | 28.88  | 30.14 | 36.04    |
> > > | MC      | 24.42  | 24.62   | 35.22    | 46.42  | 66.96 | 84.30    |
> > >
> > > Thank you again for your constructive feedback and for recognizing the contributions of our work.

---

### Official Review · Reviewer_q5B8 · 2025-07-01

**Clarity:** 3
**Significance:** 4
**Originality:** 3
**Rating:** 4
**Confidence:** 3

**Summary:**

The paper introduces Pairwise Epistemic Estimators, a new method for estimating epistemic uncertainty in regression tasks. This approach leverages an ensemble of models derived from Normalizing Flows and uses pairwise distance estimations to quantify uncertainty. The authors provide theoretical grounding, demonstrating that PaiDEs establish bounds on entropy and enhance the performance of the BALD method. Notably, PaiDEs efficiently estimate uncertainty using only a few members in ensemble, making them computationally faster than Monte Carlo-based alternatives. The effectiveness of the method is demonstrated on the active learning task on two 1D and four multidimensional regression datasets.

**Questions:**

Given that the proposed method only requires receiving a probability distribution from each ensemble member, could it be applied to other domains, such as image classification using Bayesian Neural Networks?

**Ethical Concerns:**

["NO or VERY MINOR ethics concerns only"]

**Final Justification:**

The rebuttal and follow-up discussions addressed most of my initial concerns: ensemble diversity is now clearly described; prior work and new appendix experiments justify the choice of five ensemble members, showing diminishing returns for larger ensembles; and the authors clarified that entropy-based criteria were evaluated in prior work and found ineffective in this setting.

While the addition of some extra baselines could further enrich the comparison, the current set of state-of-the-art baselines and the diversity of tasks provide sufficient empirical support. Overall, the work is technically sound, clearly presented, and addresses an important problem in uncertainty estimation for regression, with both theoretical and practical contributions. I maintain my positive assessment and recommend acceptance.

**Limitations:**

Yes

**Quality:**

3

**Strengths And Weaknesses:**

**Strengths:**
1. The paper is well-structured with clear clearly written, providing a comprehensive background on normalized flows and probabilistic network.
2. The paper offers theoretical justification for how PaiDEs establish entropy bounds using various generalized distance functions and leverage these to enhance the BALD method.
3. The authors validate the proposed epistemic uncertainty estimation method on simple 1D regression tasks, showing expected uncertainty behavior across different regions. More importantly, they demonstrate its effectiveness for active learning in multidimensional regression tasks.
4. Additionally, the paper highlights the computational efficiency of the method, achieving competitive or superior performance to Monte Carlo-based approaches with a small ensemble.



**Weaknesses:**
1. The approach for ensuring diversity of the ensemble (lines 152–153) is not clearly explained. Specifically, how is the fixed dropout mask computed? Is a different mask generated for each ensemble member? Furthermore, it is unclear for what part of the ensemble randomization or bootstrapping is applied.
2. The computational efficiency of the proposed approach appears to result from a smaller committee size, and even this small ensemble achieves competitive results. However, it remains unclear how varying the committee size impacts active learning performance. While Figure 5 presents computational time across different ensemble sizes, including active learning performance metrics for the same range would significantly strengthen the empirical analysis.
3. The paper would benefit from comparisons with additional baselines. In particular, methods based on Monte Carlo [1]. Moreover, an adaptation of least confidence method, which commonly used in classification, could be applied, given that the model outputs a distribution.
4. Minor issue: there is a typo in line 118 “wtighter” should be corrected to “tighter”.

[1] https://arxiv.org/pdf/1806.09856

---

> ### Author Rebuttal · Authors · 2025-07-30
>
> Dear Reviewer,
>
> Thank you for your thoughtful and encouraging review. We’re glad you found the paper well-structured and the background on normalizing flows and probabilistic networks helpful. We appreciate your recognition of our theoretical contributions and the empirical results—both in 1D regression and high-dimensional active learning settings.
>
> **Legend:**
> **W** – Reviewer *Weakness*
> **R** – Author *Response*
> **Q** – Reviewer *Question*
> **A** – Author *Answer*
>
> **W1:** The approach for ensuring diversity of the ensemble (lines 152–153) is not clearly explained. Specifically, how is the fixed dropout mask computed? Is a different mask generated for each ensemble member? Furthermore, it is unclear for what part of the ensemble randomization or bootstrapping is applied.
>
> **R1:** Each ensemble member is initialized with a unique random seed, trained on a bootstrapped subset of the data, and assigned a distinct dropout mask (sampled once at initialization with probability $p = 0.5$). This mask is fixed for the duration of both training and inference. These techniques—random weights, data bootstrapping, and fixed dropout—collectively ensure diversity across the ensemble. For clarity, we have added the following to the paper:
>
> > To encourage diversity in the ensemble, each member is initialized with different random weights, trained on a bootstrapped subset of the data, and assigned a fixed dropout mask sampled at the start of training (with $p = 0.5$), similar to [1]. This mask remains constant throughout training and inference.
>
> **W2:** The computational efficiency of the proposed approach appears to result from a smaller committee size, and even this small ensemble achieves competitive results. However, it remains unclear how varying the committee size impacts active learning performance. While Figure 5 presents computational time across different ensemble sizes, including active learning performance metrics for the same range would significantly strengthen the empirical analysis.
>
> **R2:** We experimented with different ensemble sizes and found that 5 components consistently provided reliable uncertainty estimates across both low- and high-dimensional tasks (see Figures 2 and 4). This is consistent with prior work [2], which also uses 5 components for normalizing flow ensembles. Our choice was thus not a limitation imposed for efficiency, but one grounded in empirical sufficiency and precedent. Additionally, other recent works [3, 4] also find that ensembles of 5 members are effective for epistemic uncertainty estimation.
>
> We also acknowledge in Figure 6 that our method becomes less effective as the number of ensemble components increases. However, this is less concerning in most deep learning applications, where ensembles are typically kept small due to computational considerations.
>
> **W3:** The paper would benefit from comparisons with additional baselines. In particular, methods based on Monte Carlo. Moreover, an adaptation of least confidence method, which is commonly used in classification, could be applied, given that the model outputs a distribution.
>
> **R3:**  Our experiments include a diverse and widely used set of baselines: BADGE, BAIT, BALD, BatchBALD, and random. These cover both uncertainty-based and diversity-driven acquisition strategies. We also evaluate across two model classes (PNEs and Nflows Base) and six environments, ranging from simple 1D tasks to high-dimensional control (Humanoid), providing a comprehensive assessment of epistemic uncertainty.
>
> MC Dropout is conceptually and practically similar to PNEs. It performs approximate Bayesian inference by applying dropout at test time, inducing a distribution over models. Each stochastic forward pass samples from this implicit posterior, analogous to how PNEs sample from an explicit weight distribution. In our setup, PNEs are instantiated by fixing different dropout masks at initialization to define each ensemble member. This approach mirrors the stochasticity of MC Dropout, while producing deterministic members that enable consistent downstream evaluation and acquisition.
>
> Recent work [2, 4, 5] has shown that deep ensembles often yield more robust and reliable uncertainty estimates than MC Dropout, particularly for regression and under distribution shift—both of which are central to our active learning setting. For example, Ovadia et al. [4] report:
>
> > Deep ensembles seem to perform the best across most metrics and be more robust to dataset shift.
>
> As for the least confidence criterion, [2] used the entropy of the output distribution as a proxy for least confidence in multi-dimensional regression, but found it performed poorly in practice. This underscores the difficulty of using such heuristics for epistemic uncertainty estimation.
>
> **W4:** Minor issue: there is a typo in line 118 “wtighter” should be corrected to “tighter”.
>
> **R4:** Thank you, this has been corrected.
>
> **Q1:** Given that the proposed method only requires receiving a probability distribution from each ensemble member, could it be applied to other domains, such as image classification using Bayesian Neural Networks?
>
> **A1:** While our method can be applied to classification tasks using ensembles, its advantages are most pronounced in regression with high-dimensional outputs. In classification, predictive entropy $H(y \mid x)$ can already be directly estimated from the categorical distribution produced by an ensemble, reducing the need for more complex estimators. In contrast, high-dimensional regression presents unique challenges for epistemic uncertainty estimation, where reliable techniques are still limited. Our approach directly addresses this gap, extending uncertainty estimation to a setting that has received far less attention than classification or 1D regression.
>
> **References:**
>
> [1] Durasov, N., Bagautdinov, T., Baque, P., & Fua, P. (2021). Masksembles for uncertainty estimation. *Proceedings of the IEEE/CVF Conference on Computer Vision and Pattern Recognition*, 13539–13548.
> [2] Berry, L., & Meger, D. (2023). Normalizing flow ensembles for rich aleatoric and epistemic uncertainty modeling. *arXiv preprint arXiv:2302.01312*.
> [3] Chua, K., Calandra, R., McAllister, R., & Levine, S. (2018). Deep reinforcement learning in a handful of trials using probabilistic dynamics models. *NeurIPS*.
> [4] Ovadia, Y., Fertig, E., Ren, J., Nado, Z., Sculley, D., Nowozin, S., ... & Snoek, J. (2019). Can you trust your model's uncertainty? Evaluating predictive uncertainty under dataset shift. *NeurIPS*.
> [5] Fort, S., Hu, H., & Lakshminarayanan, B. (2019). Deep ensembles: A loss landscape perspective. *arXiv preprint arXiv:1912.02757*.

---

> > ### Comment · Reviewer_q5B8 · 2025-08-04
> >
> > I thank the authors for their detailed rebuttal. Most of my concerns have been addressed.
> >
> > However, still several points not fully addressed:
> >
> > 1. While previous works demonstrate that ensembles with five committee members are sufficient, it would still be informative to see how the proposed approach performs with larger committee sizes.
> >
> > 2. Although a random baseline is included for comparison, incorporating additional baselines such as the least confidence criterion and entropy-based uncertainty would provide a more comprehensive analysis. These methods are computationally cheaper and illustrate a trade-off between performance and efficiency. Including them could help to demonstarte the practical benefits of the proposed approach.
> >
> > Therefore, I maintain my overall positive assessment.

---

> > > ### Author Response · Authors · 2025-08-06
> > >
> > > Dear Reviewer,
> > >
> > > Thank you for your thoughtful follow-up and for maintaining your overall positive assessment. We’re glad that most of your concerns have been addressed, and we appreciate your continued engagement.
> > >
> > > **1. Ensemble Size**
> > >
> > > We agree that evaluating different ensemble sizes could offer a more detailed empirical analysis. However, our choice of five ensemble components is supported by both prior work and consistent empirical results. Larger ensembles tend to exhibit diminishing returns, while smaller ensembles often lack the diversity needed for effective uncertainty estimation. As you noted, we do provide an analysis of how compute time scales with ensemble size for PairEpEsts in Figure 6. We believe this analysis is most relevant for practitioners applying PairEpEsts in new settings, as it helps them tailor the ensemble size to their specific computational constraints.
> > >
> > >
> > > **2. Baselines**
> > >
> > > While additional baselines can always be included, we compare PairEpEsts against five baselines: one random and four SOTA active learning approaches. Although entropy-based uncertainty is computationally cheaper due to the absence of ensembles, it consistently underperforms compared to epistemic uncertainty methods [1]. In particular, [1] shows in Figure 4 that the entropy-based criterion (brown line) performs worse in 4 out of the 6 environments we evaluate. Therefore, we believe that including it would not change the conclusions of our paper.
> > >
> > > Thank you again for your constructive suggestions.
> > >
> > > **References:**
> > >
> > > [1] Berry, L., & Meger, D. (2023). *Normalizing flow ensembles for rich aleatoric and epistemic uncertainty modeling*. arXiv:2302.01312.

---

> > > > ### Comment · Reviewer_q5B8 · 2025-08-08
> > > >
> > > > Thank you for your detailed response. I appreciate the additional discussion and the empirical justification regarding ensemble sizes and baseline comparisons. These clarifications, along with the supplementary experiments, will significantly strengthen the paper.

---

> > > > > ### Author Response · Authors · 2025-08-08
> > > > >
> > > > > Thank you for the additional comment. We have updated the manuscript to clarify that entropy-based acquisition functions were considered in prior work and found to be ineffective in our setting. Specifically, we added the following sentence:
> > > > >
> > > > > > Another potential acquisition function is the entropy of the output distribution; however, [1] showed that it was ineffective in our setting.
> > > > >
> > > > > To further justify our choice of 5 ensemble components, we have added an experiment in the appendix that analyzes the effect of ensemble size on active learning performance. This experiment was conducted on the Hopper environment, using RMSE at the final acquisition batch:
> > > > >
> > > > > **KL**
> > > > > | Ensemble Size | 3   | 4   | 5   | 7   | 10  |
> > > > > |---------------|-----|-----|-----|-----|-----|
> > > > > | RMSE          | 0.51 | 0.43 | 0.30 | 0.31 | 0.29 |
> > > > >
> > > > > **Bhatt**
> > > > > | Ensemble Size | 3   | 4   | 5   | 7   | 10  |
> > > > > |---------------|-----|-----|-----|-----|-----|
> > > > > | RMSE          | 0.50 | 0.45 | 0.29 | 0.28 | 0.30 |
> > > > >
> > > > > **References**
> > > > >
> > > > > [1] Berry, L., & Meger, D. (2023). Normalizing flow ensembles for rich aleatoric and epistemic uncertainty modeling. arXiv preprint arXiv:2302.01312.

---

### Official Review · Reviewer_5FYN · 2025-07-01

**Clarity:** 3
**Significance:** 2
**Originality:** 2
**Rating:** 5
**Confidence:** 3

**Summary:**

The paper studies an alternative to sample-based estimation of epistemic uncertainty via pairwise distance estimators. The paper derives estimators based on two measures, Bhattacharyya distance and Kullback-Leibler divergence. Across two 1d synthetic examples and high-dimensional active learning tasks, the paper shows that the proposed sample-free estimators yield improved performance at matched compute budgets.

**Questions:**

* In Figure 2, it appears that also a count-based method could perform well, at an even lower cost. I think this could be worth mentioning.
* In line 103, why does $H(\theta \mid y, x) \geq 0$ hold, given that this is a differential entropy?
* In Figure 5, for which #samples is MC evaluated here? For which #samples would MC have the same compute time as Bhatt/KL?

**Ethical Concerns:**

["NO or VERY MINOR ethics concerns only"]

**Final Justification:**

In my view this paper is a solid contribution. Though its novelty and originality is somewhat limited, it evaluates scalable and efficient methods for uncertainty estimation on high-dimensional tasks. Since the authors have addressed my remaining concerns, I increased my score to 5. Overall, I would rate this paper between 4 and 5.

**Limitations:**

yes

**Quality:**

3

**Strengths And Weaknesses:**

Strengths:
* The paper correctly identifies the computational burden of MC in most settings. Even though in many active learning settings, sample efficiency is the main bottleneck, improving compute efficiency can allow for screening more options in very high-dimensional settings.
* The experimental evaluation is solid, with a range of settings from low-dimensional to high-dimensional.

Weaknesses:
* The theoretical contribution beyond reference [35] appears limited. Nevertheless, the manuscript could benefit from a clearer outline of the novelty of the analysis.
* I am missing a comparison of MC and the proposed methods in terms of compute cost (time). For example, in Figure 3, it would be useful to indicate the #samples for MC such that the overall compute cost in terms of time is matched between MC and PaiDEs.
* While the experimental evaluation is solid, the conclusion around the benefits of sample-free estimation in high-dimensional tasks would be more significant if evaluated also on another (very) high-dimensional task beyond Humanoid. It would also be nice to see an evaluation on a common benchmark for active learning (e.g., [1]).

Typos:
* line 118: "wtighter"

[1]: Navigating the pitfalls of active learning evaluation: A systematic framework for meaningful performance assessment (https://proceedings.neurips.cc/paper_files/paper/2023/file/1ed4723f12853cbd02aecb8160f5e0c9-Paper-Conference.pdf)

---

> ### Author Rebuttal · Authors · 2025-07-30
>
> Dear Reviewer,
>
> Thank you for your thoughtful review. We appreciate your recognition that the paper correctly identifies the computational burden of MC methods, particularly in settings like active learning where epistemic uncertainty must be evaluated repeatedly across large candidate pools. While sample efficiency is often the primary focus, improving compute efficiency can dramatically expand the feasible scale of screening, especially in very high-dimensional output spaces. We're also glad you found our experimental evaluation solid. We emphasize that it covers a diverse range of tasks, two model types, and five baselines, highlighting the versatility and practical impact of our method.
>
> **Legend:**
> **W** – Reviewer *Weakness*
> **R** – Author *Response*
> **Q** – Reviewer *Question*
> **A** – Author *Answer*
>
> **W1:** The theoretical contribution beyond reference [1] appears limited. Nevertheless, the manuscript could benefit from a clearer outline of the novelty of the analysis.
>
> **R1:** We transparently cite prior work [1] for the theoretical foundations underlying PaiDEs. However, our contribution lies in how we operationalize this theory for scalable epistemic uncertainty estimation, especially in high-dimensional regression tasks where existing methods falter.
> We introduce:
> - A principled epistemic uncertainty estimator using PaiDEs.
> - A practical implementation using normalizing flows and ensembles.
> - Extensive empirical validation across six environments, two model families, and five acquisition methods.
>
> This work tackles a concrete gap in the literature: how to estimate epistemic uncertainty scalably and meaningfully for continuous, high-dimensional outputs—critical in domains like robot learning, where data collection is expensive and risk-sensitive.
> We have added the following text to clarify this:
>
> > We introduce PairEpEsts, a novel framework that leverages pairwise distance estimators [1] for epistemic uncertainty in deep ensembles with probabilistic outputs—focusing on high-dimensional regression tasks previously underserved by the literature (see Section 4).
>
> **W2:** I am missing a comparison of MC and the proposed methods in terms of compute cost (time). For example, in Figure 3, it would be useful to indicate the #samples for MC such that the overall compute cost in terms of time is matched between MC and PaiDEs.
>
> **R2:** We have clarified this in Figure 3. MC matches the compute time of KL and Bhatt estimators when using approximately 10 samples. This now appears in the figure.
>
> **W3:** While the experimental evaluation is solid, the conclusion around the benefits of sample-free estimation in high-dimensional tasks would be more significant if evaluated also on another (very) high-dimensional task beyond Humanoid. It would also be nice to see an evaluation on a common benchmark for active learning (e.g., [2]).
>
> **R3:** We chose high-dimensional regression environments from the MuJoCo suite (Ant, Humanoid) to explicitly address a gap in the literature: most benchmarks, including [2], focus on classification or 1D regression, which do not reflect the challenges of real-world control tasks. Ant and Humanoid environments test uncertainty estimation in more realistic, complex settings with structured, high-dimensional outputs. While we do include 1D toy examples—similar to prior work—our primary goal was to evaluate performance in more demanding settings. Although our approach builds on [3], its extension to these control domains is not straightforward; it requires adapting to the computational challenges of continuous, high-dimensional spaces. Our work provides this bridge. Applying our method to even higher-dimensional problems like image generation is a promising direction, though it demands additional architecture-specific considerations, which we leave for future work.
>
> **Q1:** In Figure 2, it appears that also a count-based method could perform well, at an even lower cost. I think this could be worth mentioning.
>
> **A1:** We agree that in the 1D toy setting of Figure 2, a simple count-based method could serve as an estimate for epistemic uncertainty and potentially perform well at low computational cost. However, count-based methods generally fail to scale to higher-dimensional input spaces such as those in Hopper, Ant, and Humanoid. In these settings, the input space becomes exponentially larger and sparser, making it difficult to obtain meaningful counts. Moreover, count-based methods lack the ability to leverage the learned representations of the model, which are essential in high-dimensional and complex tasks. For these reasons, more principled estimators, such as those we propose, are necessary to provide reliable uncertainty estimates in realistic settings.
> We have added the following text to the paper to make this clear:
>
> > While count-based methods may suffice in low-dimensional settings, they do not generalize well to high-dimensional spaces where data is sparse and frequency-based heuristics fail. Our approach leverages learned representations, enabling uncertainty estimation in complex, high-dimensional scenarios.
>
> **Q2:**  In line 103, why does $H(\theta|y,x) \geq 0$ hold, given that this is a differential entropy?
>
> **A2:** $\theta$ denotes the discrete index of ensemble components (we use 5), making $H(\theta|x,y)$ a Shannon entropy, not differential entropy. This ensures $H(\theta|x,y) \geq 0$. We’ve clarified this in the text:
>
> > The upper bound follows from Equation 3, and $H(\theta|x,y) \geq 0$ since $\theta$ is modeled as a discrete random variable (ensemble index), unlike $x$ and $y$ which are continuous.
>
> **Q3:** In Figure 5, for which #samples is MC evaluated here? For which #samples would MC have the same compute time as Bhatt/KL?
>
> **A3:** The MC results in Figure 5 use 5000 samples per acquisition candidate across all environments, consistent with the setup in Figure 4. We have updated the caption of Figure 5 to explicitly clarify this.
>
> The number of samples for MC to match compute time with Bhatt/KL varies by environment due to dimensionality. We’ve added a table to the paper summarizing these values:
>
> | Environment | Dimensionality | MC Samples (Runtime Matched) |
> |-----------|------------------------|--------------------------|
> | Hetero    | 1 dim                  | 500                      |
> | Bimodal   | 1 dim                  | 500                      |
> | Pendulum  | 3 dims                 | 100                      |
> | Hopper    | 11 dims                | 10                       |
> | Ant       | 27 dims                | 5                        |
> | Humanoid  | 257 dims               | 5                        |
>
> As noted in Section 5.3, we use 100 acquisition candidates for MC in the Humanoid environment, compared to 1000 in other tasks, due to computational constraints. This reduced set allows Monte Carlo to match the performance of Bhatt and KL methods.
>
> This analysis has been added to the paper.
>
> **Typo:** Line 118: "wtighter"
>
> **R:** Thank you, this has been corrected.
>
> **References**
>
> [1] Kolchinsky, A., & Tracey, B. D. (2017). Estimating mixture entropy with pairwise distances. *Entropy*, 19(7), 361.
> [2] Lüth, C., Bungert, T., Klein, L., & Jaeger, P. (2023). Navigating the pitfalls of active learning evaluation: A systematic framework for meaningful performance assessment. *Advances in Neural Information Processing Systems*, 36, 9789-9836.
> [3] Berry, L., & Meger, D. (2023). Normalizing flow ensembles for rich aleatoric and epistemic uncertainty modeling. arXiv:2302.01312.

---

### Official Review · Reviewer_3iVP · 2025-07-02

**Clarity:** 3
**Significance:** 2
**Originality:** 2
**Rating:** 4
**Confidence:** 3

**Summary:**

The work proposes a novel estimator for epistemic uncertainty estimation in ensemble models for regression tasks, which does not require Monte Carlo sampling. The estimator (Pairwise Epistemic Estimators) is based on Pairwise-Distance Estimators (PaiDEs) to obtain a lower/upper bound for the mutual information (or equivalently, the epistemic uncertainty). The PaiDEs require closed-form expressions for the divergences/distances between distributions, which motivates the authors to use Normalizing Flows (NFs) to model the distribution and reduce the target distributions' premetrics to the base distributions. The algorithm is validated by active learning experiments compared to other uncertainty-based baselines.

**Questions:**

1. Have you ever tried other active learning benchmark algorithms? (See Weaknesses point 3)

2. How precise is your evaluation of epistemic uncertainty (as a final output rather than an intermediate product)?

3. Typos: line 118 "wtighter" should be "tighter"?

**Ethical Concerns:**

["NO or VERY MINOR ethics concerns only"]

**Final Justification:**

I think this paper gives a good sampling-free way of estimating the epistemic uncertainty (though with many limitations to its applicability). The theoretical part is largely due to previous works (which limits the originality), but the application of the theory is worth noticing. Therefore, I will raise my score to 4.

**Limitations:**

Yes

**Quality:**

3

**Strengths And Weaknesses:**

Strengths:

1. The proposed method is sampling-free, which will boost the computation once we are in a high-dimensional space.

2. The theoretical understanding of the algorithm is solid.

3. The empirical evidence is impressive.

4. The presentation is clear and easy to follow.

Weaknesses:

1. The method only applies when all premetrics are of closed-form expressions. This limits the applicability in many scenarios. Also, the numerical experiments are all conducted in Gaussian bases. It remains unclear how robust it is with skewed or heavy-tailed distributions.

2. The estimator of the epistemic uncertainty is biased (e.g., either a lower bound or an upper bound in the two examples). It shall be fine if the uncertainty is only an intermediate product (e.g., uncertainty-based active learning), but if one wants the epistemic uncertainty estimator as the final output, it becomes a big problem. The paper also lacks an analysis of the potential bias.

3. The active learning algorithm benchmark is restricted. Uncertainty-based query strategy is only one category of active learning algorithms; it remains unclear if they are SOTA in many scenarios. More comparisons (e.g., coverage/representation-based active learning algorithms) should be included. (update: solved)

4. The theoretical contribution of this paper is limited. All the theory parts are from the previous PaiDE paper.

---

> ### Author Rebuttal · Authors · 2025-07-30
>
> Dear Reviewer,
>
> Thank you for your thoughtful review and for recognizing the strengths of our work. We appreciate your acknowledgment that our proposed method is sampling-free, offering a significant computational advantage. We're also glad you found the theoretical foundations of our approach to be solid and the empirical results compelling.
>
> **Legend:**
> **W** – Reviewer *Weakness*
> **R** – Author *Response*
> **Q** – Reviewer *Question*
> **A** – Author *Answer*
>
> **W1:** The method only applies when all premetrics are of closed-form expressions. This limits the applicability in many scenarios.
>
> **R1:** We agree this is an important consideration and note it explicitly in Section 4. However, PairEpEsts are broadly applicable, including to non-parametric models such as normalizing flows, which capture complex, multimodal, and skewed distributions. This is because PairEpEsts can operate on latent distributions such as the tractable base distributions in normalizing flows where closed-form premetrics are available.
>
> PairEpEsts are also flexible in their choice of distributional distance measures. They support a wide range of options, such as KL divergence, Bhattacharyya distance, Hellinger distance, total variation distance, and Wasserstein distance, provided these measures admit closed-form or efficiently computable expressions. Many commonly used models, including those that produce outputs in the exponential family (e.g., Gaussian, Bernoulli, Poisson, Dirichlet), as well as models with tractable latent distributions, satisfy this condition.
>
> In summary, while PairEpEsts rely on closed-form premetrics, they can be applied to a broad class of predictive models, making them well-suited for a variety of machine learning applications.
>
> **W2:** The numerical experiments are all conducted in Gaussian bases. It remains unclear how robust it is with skewed or heavy-tailed distributions.
>
> **R2:** Our choice of Gaussian bases aligns with standard practice in robotic control tasks and many other fields, where they’ve proven to be highly effective. This also ensures more comparability to prior work.
>
> That said, we acknowledge the importance of modeling heavy-tailed and skewed predictive distributions in other domains such as finance and insurance. PairEpEsts are not inherently limited to Gaussian bases. They are compatible with skewed or heavy-tailed base distributions—so long as the pairwise premetric remains tractable. Recent work [1, 2] introduces tail-adaptive normalizing flows specifically designed for such heavy-tailed distributions. We’ve clarified this in the updated manuscript with the following addition:
>
> > PairEpEsts naturally extend to skewed or heavy-tailed base distributions, as long as the pairwise premetric admits a closed-form expression. Tail-adaptive normalizing flows [1, 2] offer a promising direction for incorporating heavy-tailed behavior into normalizing flows. Exploring their integration with PairEpEsts remains an exciting avenue for future work, especially in domains where tail risk is critical.
>
> **W3:** The estimator of the epistemic uncertainty is biased (e.g., either a lower bound or an upper bound in the two examples). The paper also lacks an analysis of the potential bias.
>
> **R3:** Unlike aleatoric uncertainty, there is no universally accepted ground truth for epistemic uncertainty. However, it is agreed that inputs with fewer observations should exhibit higher epistemic uncertainty, as in Figure 2. In this context, biased metrics for epistemic uncertainty estimation have become a common tool [3, 4].
>
> To further assess whether our methods introduce reordering in the estimated uncertainties, we computed Spearman’s rank correlation coefficient, a non-parametric statistic that measures the strength and direction of the monotonic relationship between two ranked variables. Specifically, we compared the rankings produced by our estimators (KL and Bhatt) against those from MC estimation on the data used in Figure 8. A coefficient of 1 indicates perfect agreement in ranking, 0 indicates no correlation, and -1 indicates complete inverse ranking.
>
> As shown below, the correlation coefficients are all very close to 1, suggesting that our estimators preserve the relative ordering of uncertainty values nearly perfectly. This implies that any bias introduced by our method has minimal impact. Moreover, all p-values are extremely small (largest p = 1.11e-83), confirming the statistical significance of the correlations.
>
> This analysis has been added to the paper.
>
> | Model       | Dataset | KL     | Bhatt  |
> |-------------|---------|--------|--------|
> | NFlows Base | Bimodal | 0.9958 | 0.9958 |
> |             | Hetero  | 0.9976 | 0.9986 |
> | PENs        | Bimodal | 0.9893 | 0.9893 |
> |             | Hetero  | 0.9943 | 0.9972 |
>
> **W4:** The active learning algorithm benchmark is restricted.
>
> **R4:** We include BADGE, BAIT, and BatchBALD, three widely used and highly cited active learning methods that incorporate representation and diversity-based strategies:
>
> - **BADGE**: Combines feature space diversity via gradient embeddings.
> - **BAIT**: Utilizes influence functions to promote diversity-aware selection.
> - **BatchBALD**: Combines uncertainty with feature space diversity.
>
> These methods are not purely uncertainty-based; rather, they are varied approaches that represent state-of-the-art acquisition strategies beyond uncertainty alone. We believe this selection strikes a strong balance between methodological relevance, algorithmic diversity, state-of-the-art, and computational feasibility in the active learning setting.
>
> **W5:** The theoretical contribution of this paper is limited. All the theory parts are from the previous PaiDE paper.
>
> **R5:** We transparently cite prior work [5] for the theoretical foundations underlying PaiDEs. However, our contribution lies in how we operationalize this theory for scalable epistemic uncertainty estimation, especially in high-dimensional regression tasks where existing methods falter.
>
> We introduce:
>
> - A principled epistemic uncertainty estimator using PaiDEs.
> - A practical implementation using normalizing flows and PNEs.
> - Extensive empirical validation across six environments, two model families, and five acquisition methods.
>
> This work tackles a concrete gap in the literature: how to estimate epistemic uncertainty scalably and meaningfully for continuous, high-dimensional outputs—critical in domains like robot learning, where data collection is expensive and risk-sensitive.
>
> **Q1:** Have you ever tried other active learning benchmark algorithms?
>
> **A1:** We include BADGE, BAIT, and BatchBALD, which incorporate both uncertainty and diversity. See **R4** above.
>
> **Q2:** How precise is your evaluation of epistemic uncertainty (as a final output rather than an intermediate product)?
>
> **A2:** While epistemic uncertainty is used here primarily as an acquisition signal, we also provide qualitative visualizations in 1D settings (Figure 8) to assess estimator behavior across the input space. These plots show that PairEpEsts effectively capture key structural patterns in uncertainty, assigning higher uncertainty to under-sampled regions as expected. See **R3** above.
>
> **Q3:** Typo on line 118 ("wtighter")
>
> **A3:** Thank you for spotting this typo. It has been corrected.
>
> **References**
>
> [1] Laszkiewicz, M., Lederer, J., & Fischer, A. (2022). *Marginal tail-adaptive normalizing flows*. ICML.
> [2] Hickling, T., & Prangle, D. (2024). *Flexible Tails for Normalizing Flows*. arXiv:2406.16971
> [3] Malinin, A., & Gales, M. (2020). *Uncertainty estimation in autoregressive structured prediction*. arXiv:2002.07650
> [4] Schweighofer, K. et al. (2023). *An improved information-theoretic measure of predictive uncertainty*. arXiv:2311.08309
> [5] Kolchinsky, A., & Tracey, B. (2017). *Estimating mixture entropy with pairwise distances*. *Entropy*, 19(7), 361.

---

> > ### Comment · Reviewer_3iVP · 2025-08-06
> >
> > Thank you for your detailed response. I think my major concerns have been mostly addressed. I have raised my score from 3 to 4.

---

> > > ### Author Response · Authors · 2025-08-06
> > >
> > > Dear Reviewer,
> > >
> > > Thank you for your follow-up and for increasing your score. We're glad to hear that your major concerns have been mostly addressed.

---

### Author Response · Authors · 2025-08-08

Dear Reviewers and Area Chair,

As the discussion period comes to a close, we would like to sincerely thank all reviewers and AC for their engagement and hard work throughout this process. We’re pleased that we were able to address most of the concerns raised, and we’d like to briefly highlight two key points that were mentioned by multiple reviewers:


**1. Lack of theoretical results**

The theoretical foundations of our method build on prior work [1] on PaiDEs, which is cited and referenced throughout the manuscript. Our primary contribution lies in operationalizing this theory for scalable epistemic uncertainty estimation, particularly in high-dimensional regression tasks where many existing approaches struggle.

Specifically, we introduce:

- A principled epistemic uncertainty estimator based on PaiDEs,
- A practical implementation leveraging normalizing flows and ensembles, and
- Extensive empirical validation across six environments, two model families, and five acquisition strategies.

Additionally, during the review process, we included a new theoretical result unique to this work: we prove that PairEpEst-KL is a tighter upper bound on mutual information than EPKL. Please refer to **A1** in our response to reviewer **yg5h** for details.


**2. Bias introduced by our method**

While PairEpEsts introduce a biased estimator of mutual information, this is not a significant issue in the context of epistemic uncertainty, which lacks ground-truth values (unlike aleatoric uncertainty). Within the community, epistemic uncertainty is commonly interpreted as being higher for points with fewer observations. To support this, we included an experiment demonstrating that our estimator maintains this desirable property. Please see our response **R3** to reviewer **3iVP**.

Thank you again for your time and for helping us improve our manuscript.

**References**

[1] Kolchinsky, A., & Tracey, B. D. (2017). *Estimating mixture entropy with pairwise distances*. Entropy, 19(7), 361.

---

### Decision · Program_Chairs · 2025-09-17

**Decision:**

Accept (poster)

**Comment:**

This paper is about the use of pairwise distance estimators in ensemble models for epistemic uncertainty estimation, which can be computed without performing sampling, providing computational advantages versus other standard methods that require sampling.

Reviewers appreciated the proposed method being sampling-free, good experimental evaluation and theoretical justifications. The negatives are the limited set of baselines and comparison using more complex datasets.

There was a good amount of discussion and all reviewers vote to accept. This paper makes a good contribution of providing a uncertainty estimation method that does not require sampling, being much faster than alternative methods, which is a key weakness of uncertainty estimation methods and improving computational complexity is required for real-world application use.

I recommend that the authors consider the feedback from reviewers in their final version, in particular from reviewers 5FYN about pitfalls in evaluating active learning, and comparisons with standard baselines like MC-Dropout (Reviewer q5B8), clarify some figures (Reviewer yg5h) and more complex datasets, which can provide more information for future research.